# Prediction and characterisation of the human B cell response to a heterologous two-dose Ebola vaccine

Daniel O'Connor [1,2,5] ✉, Elizabeth A. Clutterbuck [1,2,5], Malick M. Gibani [3], Sagida Bibi[1,2], Katherine A. Sanders[1,2], Rebecca Makinson [2,4], Dominic F. Kelly[1,2] & Andrew J. Pollard [1,2]

Ebola virus disease (EVD) outbreaks are increasing, posing significant threats to affected communities. Effective outbreak management depends on protecting frontline health workers, a key focus of EVD vaccination strategies. IgG specific to the viral glycoprotein serves as the correlate of protection for recent vaccine licensures. Using advanced cellular and transcriptomic analyses, we examined B cell responses to the Ad26.ZEBOV, MVA-BN-Filo EVD vaccine. Our findings reveal robust plasma cell and lasting B cell memory responses post-vaccination. Machine-learning models trained on blood gene expression predicted antibody response magnitude. Notably, we identified a unique B cell receptor CDRH3 sequence post-vaccination resembling known *Orthoebolavirus zairense* (EBOV) glycoprotein-binding antibodies. Single-cell analyses further detailed changes in plasma cell frequency, subclass usage, and CDRH3 properties. These results highlight the predictive power of early immune responses, captured through systems immunology, in shaping vaccine-induced B cell immunity.

Ebola virus disease (EVD) remains a feared and serious threat with epidemic potential. Periodic outbreaks—predominantly affecting Central and West African populations—are associated with high mortality rates, alongside severe challenges to healthcare infrastructure and the economy of affected countries. Non-pharmaceutical interventions are the mainstay of epidemic management. Rapid development of novel vaccines was initiated in response to the West African Epidemic between 2014 and 2016 of the *Orthoebolavirus zairense* (EBOV), and the principal target antigen for vaccine-induced immune response is the surface glycoprotein (GP). Two vaccines have been licensed by European Medicines Agency (EMA) for individuals aged 1 year and older to protect against EVD (https://www.ema.europa.eu). The first World Health Organization prequalified vaccine was the live-attenuated rVSVΔG-ZEBOV-GP (ERVEBO, Merck), which demonstrated safety, immunogenicity and efficacy in a ring-vaccination study[1]. A second vaccine Ad26.ZEBOV (Zabdeno, Janssen) used with MVA-BN®-Filo (Mvabea, Janssen) has demonstrated safety and immunogenicity in healthy volunteers, including down to 1 years of age, with evidence for efficacy being inferred from immunobridging studies[2–4].

A phase 1 clinical trial (EBL1001) in the UK compared dosing order and spacing of two vaccines: Ad26.ZEBOV (adenovirus serogroup 26 vaccine encoding EBOV Mayinga GP, dose 1) and MVA-BN-Filo (modified vaccinia Ankara encoding GPs from EBOV, *Orthoebolavirus sudanense* (SUDV), *Orthomarburgvirus marburgense* (MARV), and the *Orthoebolavirus taiense* (TAFV) nucleoprotein, dose 2). In this trial, 97% of participants receiving Ad26.ZEBOV as dose 1 had detectable EBOV-GP IgG responses by day 28, and all participants had detectable IgG responses 21 days post-boost, with responses still detectable after 8 months[5]. The phase 2 trial (EBL2001, EVOLVE) conducted in Oxford and London, UK, involved a two-dose regimen with Ad26.ZEBOV (dose

[1]Oxford Vaccine Group, Department of Paediatrics, University of Oxford, Oxford, UK. [2]NIHR Oxford Biomedical Research Centre, Oxford, UK. [3]Department of Infectious Disease, Imperial College London, St Mary's Campus, London, United Kingdom. [4]The Jenner Institute, Nuffield Department of Medicine, University of Oxford, Oxford, UK. [5]These authors contributed equally: Daniel O'Connor, Elizabeth A. Clutterbuck. ✉e-mail: daniel.oconnor@paediatrics.ox.ac.uk

1) and MVA-BN-Filo (dose 2). Participants were randomized into three groups to receive dose 2 at either day 29, 57 or 85 post dose 1. The spacing of the doses was compared to determine the optimal spacing for enhancing the immunogenicity outcomes and to have some flexibility of dosing in the field. The trial reported outcomes of safety and immunogenicity, showing that the dosing regimens were equally immunogenic and that increasing the dosing interval enhanced initial antibody responses, with elevated serum antibody levels at day 365. CD4+ and CD8 + T cell responses were observed in 37%–48% (CD4) and 55%–67% (CD8) of participants at day 21 post dose 2, with CD8 responses maintained in 42%–56% of participants at day 365[6]. A similar robust outcome in serum antibody, CD4+, and CD8 + T cell responses was observed in the EBL2002 trial, conducted at sites in Kenya, Uganda, and Burkina Faso, using the same trial design[7]. Long-term follow-up of antibody and T cell responses from the UK EBL2001 study was undertaken as part of the PRISM study (under review), which looks at the maintenance of immunity four years after dose 2.

Although the mechanistic correlates of vaccine-induced protection against EVD have not been identified, preclinical and phase 2/3 clinical studies have suggested that levels of IgG specific to the viral GP best correlate with protection[8–10]. An immunobridging study using a non-human primate (NHP) model was used to infer protective effects of EBOV-GP binding antibody levels obtained from clinical trials and demonstrated protection following the two-dose vaccine regimen (Ad26.ZEBOV, MVA-BN-Filo)[3]. Vaccine-induced immune memory has been proposed as a correlate of protection, as the anamnestic response can potentially halt disease progression during the extended viral incubation period[11]. Rapid recall responses have been described after a Ad26.ZEBOV booster vaccination of individuals who received a primary Ad26.ZEBOV, MVA-BN-Filo vaccine regimen[12].

In the present study, we perform detailed assessment of the B cell responses underlying the humoral immunity induced by this heterologous two-dose EVD vaccine regimen—using contemporary high-resolution transcriptomic and cellular phenotyping. This clinical study provides insight into the generation and maintenance of the long-lasting B cell immunity following EVD vaccination of humans.

## Results

### Ad26.ZEBOV followed by MVA-BN-Filo vaccination induces persistent EBOV-GP specific B cell memory responses

Within the EBL2001 and EBL2002 trials, participants were recruited in Cohorts (Supplementary Fig. 1). For EBL2001, in the UK, a sentinel Exploratory Cohort 1 ($n = 30$) was recruited specifically to assess peak plasma cell time points, following dose 1 and dose 2, to target sample collection in Cohorts 2. Cohort 2 was the exploratory cohort for collection of PBMCs and DNA samples, while Cohort 3 (not discussed in this manuscript) was for main clinical trial objectives only[6]. Each study Cohort was randomised into Group 1, Group 2 or Group 3 depending on when they would receive the MVA-BN-Filo (dose 2). For EBL2002 (Kenya, Burkina Faso and Uganda), samples were only available from Cohort 2 (vaccine Groups 1 and 2). Sampling timepoints and vaccination regimens are shown in Supplementary Fig. 1, for each Study, Cohort and Group, along with timepoints tested for each analysis. In the UK trial, EBL2001, participants were recruited from the Oxford area. Cohort 1, consisted of ($n = 30$) healthy adults aged 18–65 years old, and was recruited specifically, with a more intense sample schedule, for determination of the optimal plasma cell timing over 14 days post dose 1 and 9 days post dose 2. Cohort 2 ($n = 50$ healthy adults aged 18–65 years) was recruited for immunogenicity, along with exploratory analysis of the B cell response via ELISpot, flow cytometry and transcriptomics, and included long term follow-up to day 265. A further study, PRISM, recalled participants from Cohort 2 at four years post MVA-BN-Filo (V1), to look for long-term maintenance of immune memory (manuscript in preparation). For memory B cell responses, six

participants in each vaccine group (G1, G2 and G3) had PBMCs available at V1 (4 years post dose 2) and V2 (4.5 year post dose 2). For analysis of BMEM responses in none-UK participants, PBMC samples were obtained from the EBL2002 trial sites in Kenya, Uganda and Burkina Faso. Only PBMCs from participants in vaccine Group 1 ($n = 8$) and Group 2 ($n = 11$) were available.

In the EBL2001 Study, B cell memory (BMEM) responses were observed following a single dose of Ad26.ZEBOV (dose 1), Fig. 1 and the magnitude of the response (fold change) was greater with the longer interval (d85, group 3, $p = 0.0111$, Fig. 1a, b and Supplementary Table 3). A fold change in BMEM was not observed after Ad26.ZEBOV (dose 1) in the EBL2002 Study (dotted lines, Fig. 1a, b and Supplementary Table 5). Administration of MVA-BN-Filo (dose 2) in EBL2001 on d29, d57 or d85 enhanced the BMEM frequency, resulting in a significant increase above baseline at d21 post MVA (G1 d50, $p = 0.0029$; G2, d78, $p = 0.0065$; G3 d106, $p = <0.0001$, Supplementary Table 3). This was also observed in EBL2002 when MVA-BN-Filo was administered at d29 or d57 (Fig. 1a and Supplementary Table 5). The fold change in BMEM following MVA-BN-Filo was lower than after Ad26.ZEBOV in the EBL2001 cohort but greater in the EBL2002 cohort (Fig. 1c). The minimum to maximum ranges of EBOV-GP specific BMEM/million cultured PBMCs in EBL2001 vs EBL2002 were (Group 1: 0.10–42.50 vs 0.10–21.25) and (Group 2: 6.75–153.8 vs 0.10–10.10). The frequency of Total IgG-BMEM in all studies was similar across all Study Groups at all time points (Supplementary Fig. 15a).

In the EBL2001, the frequency of IgG-BMEM was maintained, significantly above baseline, at six months post MVA (dose 2, G1, d209, $p = <0.0001$; G2, d237, $p = 0.0005$ and G3, d265, $p = 0.0111$) (Fig. 1d and Supplementary Table 3). In the EBL2002 Study, the fold rise after MVA (dose 2), was maintained at d180 and d365, and was higher than that observed in the EBL2001 Study (Fig. 1d). No difference between study country was observed in EBL2002 (Burkina Faso, Kenya or Uganda, Supplementary Fig. 16b). In the follow-up study (PRISM), Cohort 2 participants from EBL2001 were recruited to return, 4 years post dose 2, to assess long term maintenance of immunity. For assessment of BMEM frequencies, six volunteers within each of Groups 1, 2 and 3 returned. These were participants had BMEM data available from the original study so that all time points could be paired for analysis. The follow-up results are shown in Fig. 1a and Supplementary Table 4, where at V1 (4 years post dose 2) and V2 (a further six months later), there appeared to be minimal waning in the frequency of circulating, EBOV-GP specific IgG-BMEM.

### Ad26.ZEBOV followed by MVA-BN-Filo induces a robust plasma cell response to the Zaire GP

Total antibody (IgG) secreting plasma cells (IgG-ASC) and EBOV-GP specific IgG-ASC (EBOV-GP-ASC) were enumerated by ex vivo ELISpot at sequential days following Ad26.ZEBOV (dose 1) and MVA-BN-Filo (dose 2) as part of the EBL2001 Study. For Cohort 1, the time points post dose 1 and dose 2 were chosen to capture the peak of vaccine-specific plasma cells in peripheral blood (Fig. 1e). The Total-IgG-ASC frequency peaked at d9–d11 post dose 1 (Ad26.ZEBOV) and d5-d7 post dose 2 (MVA-BN-Filo) (Fig. 1e and Supplementary Table 1). The proportion of EBOV-GP-ASC during the first two weeks after the Ad26.ZEBOV, peaked at d11 (Fig. 1e and Supplementary Table 1), at d11–13 the response was 0.6%–1.3% of the total-IgG-ASC in circulation.

Following MVA-BN-Filo boosting (dose 2), an increase in EBOV-GP-ASC frequency was observed by day 7 (Fig. 1e and Supplementary Table 1), reaching significance in all groups (G1-d36, $p = 0.0305$, G2-d64, $p = <0.0001$ and G3-d92, $p = <0.0001$). In Groups 2 and 3, receiving dose 2 at either d57 or d85, the EBOV-GP-ASC response remained significantly elevated above baselines at day 9 post dose 2 (G2-d66, $p = 0.0011$ and G3-d94, $p = <0.0001$, Fig. 1e and Supplementary Table 1). These results informed a reduced time point schedule for Cohort 2 of day 1 and d11, post dose 1 and d0 and d7 post dose 2 for

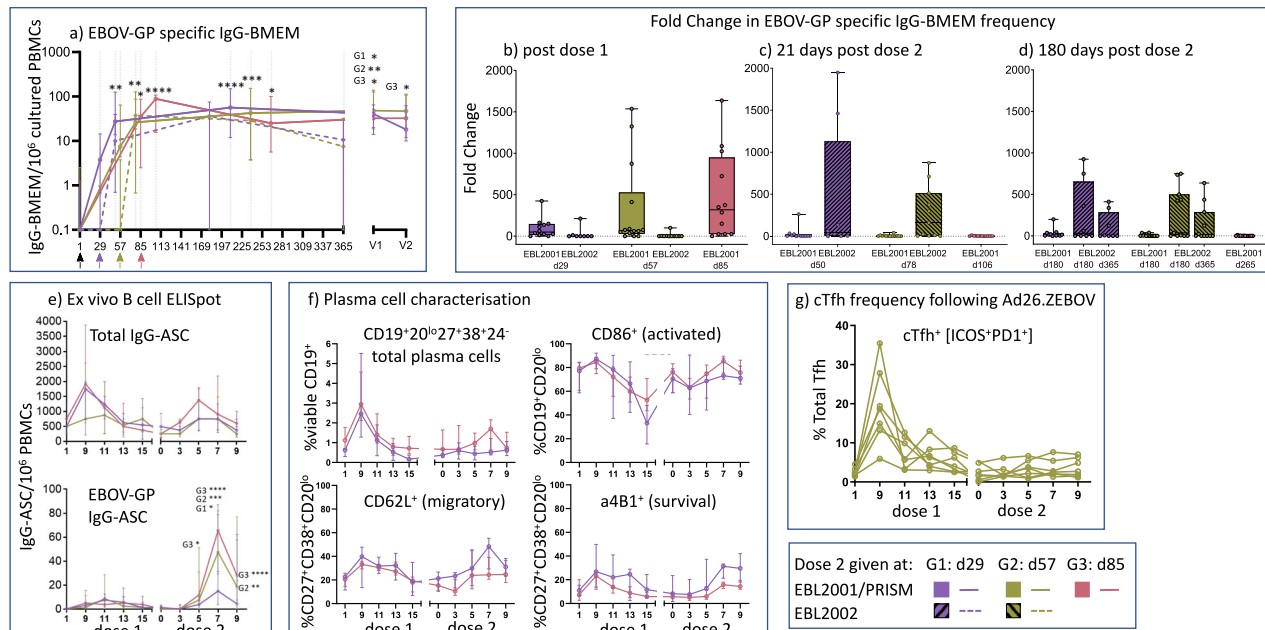

**Fig. 1 | The EBOV-GP specific IgG-BMEM response induced by Ad26.ZEBOV (dose 1) and boosted by MVA-BN-Filo (dose 2) given at d29 (Group 1, purple), d57 (Group 2, olive) or d85 (Group 3, coral). a** IgG-BMEM ELISpot response: Arrows indicate the time of prime (black) and boost coloured by group. Participants recruited as part of the EBL2001 (UK study) responses were measured at d1, d29, d50, d237 (Gr 1 (purple, solid line); d1, d57, d78, d237 (Grp2, olive, solid line) and d1, d85, d106, d265 (Grp 3, coral, solid line). Similarly, participants followed up four years later as part of the UK PRISM study (at V1 and V2) are also displayed. Also included in this figure are the responses of participants recruited as part of EBL2002 (Group 1 (purple, dotted lines, d1, d29, d50, d185, d365) and Group 2 (golive, dotted lines, d1, d57, d78, d185, d365). Data are expressed as the median (+/− IQR) of IgG-BMEM/million PBMCs. ANOVA, two-sided Freidmans test, with Dunn's multiple comparison test, p-values *<0.05, **<0.01, ***, 0.001, ****<0.0001 shown for EBL2001 and PRISM only. **b** Fold Change in IgG-BMEM at d29/d57/d85 post dose 1. **c** Fold Change in IgG-BMEM at day 21 post MVA-BN-Filo (dose 2). **d** Fold Change at d180 post dose 2. Median bar is shown and error bars are

minimum and maximum values for (**b–d**). **e** Ex vivo ELISpot Frequency of Total IgG-ASC, (top) and EBOV-GP specific IgG-ASC, (bottom), following dose 1 (d1, d9, d11, d13, d15) and dose 2 (administered at d29, purple line); d57, olive line; or d85, coral line), with frequency quantified on d0, 3, 5, 7 and 9 post dose 2. Data are expressed as Median (+/− IQR). Two-sided one-way ANOVA, with Dunn's multiple comparison test, p-values *<0.05, **<0.01, ***, 0.001, ****<0.0001. **f** Flow cytometric characterisation of plasma cells (CD19 + CD20loCD38 + CD27 + ) expressing CD86, CD62L or a4β1, (the gating strategy is shown in Supplementary Figs. 17, 18 and 19)., Date are presented as percentage of total viable CD19 + B cells, or %CD27 + CD38+CD20lo plasma cells in PBMCs obtained at d1, d9, d11, d13 and d15 post dose 1 and d0, d3, d5, d7 and d9 post dose 2 following MVA-BN-Filo at d29 (Group 1, purple) or d85 (Group 3, coral). Data are expressed as the median (+/− IQR). **g** Activated (ICOS$^+$PD1$^+$) cTfh response following Ad2.ZEBOV as dose 1 (d1, d9, d11, d13, d15) and MVA-BN-Filo administered at d57 post dose 1 (Group 2), with samples obtained on d0, d3, d5, d7 and d9 post dose 2. Gating strategies are shown in Supplementary Fig. 16c.

plasma cell analysis, with statistically significant increases in EBOV-GP-ASC by day 11 in all groups (p = <0.01, Supplementary Fig. 16d and Supplementary Table 2) and statistically significant increases by d7 post dose 2 (p = <0.0001 in all groups, Supplementary Fig. 16d and Supplementary Table 2).

#### Activated and migratory plasma cells were seen following Ad26.ZEBOV and MVA-BN-Filo

Here, we show the phenotypic characterisation of peripheral blood plasma cells from EBL2001 Cohort 1 participants (Group 1 dose 2-d29, purple and Group 3 dose 2-d85, coral) as shown in Fig. 1f and Supplementary Figs. 17–20. The CD19$^+$CD20$^{lo}$CD27$^+$CD24$^-$CD38$^+$ total blood plasma cell response (Fig. 1f, Top), showed the same kinetic as that of the ELISpot Total-IgG-ASC response (Fig. 1e, Top). These plasma cells were also predominantly class switched (IgD-IgM-) plasma cells (Supplementary Fig. 17m, dotted lines). Plasma cells expressing migration marker, CD62L$^+$ and survival marker, α4β1$^+$ (Fig. 1f and Supplementary Fig. 18k, m), showed a circulatory kinetic similar to the EBOV-GP specific ASC ELISpot response (Fig. 1e, bottom panel). The presence of activated CD86$^+$ plasma cells peaked at day 9 post dose 1 and then at d7 post dose 2 (Fig. 1f and Supplementary Fig. 18i), also following the same kinetic as the ELISpot EBOV-GP specific response. While these findings are not statistically significant, they do point to a trend of activated, migratory plasma cell responses to both dose 1 and dose 2 administration.

#### Ad26.ZEBOV prime induces robust activated cTfh response

Phenotypic analysis of circulating T-follicular helper (cTfh) cells was undertaken on samples obtained in the first two weeks following Ad26.ZEBOV and 1 week post MVA-BN-Filo. The proportion of activated (ICOS$^+$ PD1$^+$) cTfh cells were analysed in a total of seven, EBL2001 Cohort 1 participants recruited to Group 2 (MVA-BN-Filo given on day 57). ICOS$^+$PD1$^+$ cTfh cells were detected in all participants and their frequency of total cTfh cells peaked at day 9 post Ad26.ZEBOV administration and disappeared rapidly by day 13 (Fig. 1g and Supplementary Fig. 16c). The proportion of ICOS$^+$ PD1$^+$ cTfh cells did not show any increase up to 9 days following the MVA-BN-Filo vaccine (Fig. 1g).

#### Extensive differential gene regulation coinciding with the peak of plasma cell responses following Ad26.ZEBOV and MVA-BN-Filo vaccines

We next explored gene regulation with bulk RNA-sequencing of whole blood samples taken at the peak of plasma cell responses. We observed 25 differential expressed genes (DEGs, FDR < 0.05) 10 days after the first dose of study vaccine (Ad26.ZEBOV) and 442 DEGs 7 days after the second dose of vaccine (MVA-BN-Filo) (Fig. 2a, b). General agreement was seen in terms of the direction of gene regulation—at the peak of plasma cell responses—following both study vaccines (Fig. 2c). Sixteen genes were found to be differentially expressed following both vaccines—the majority of which encoded immunoglobulin gene segments

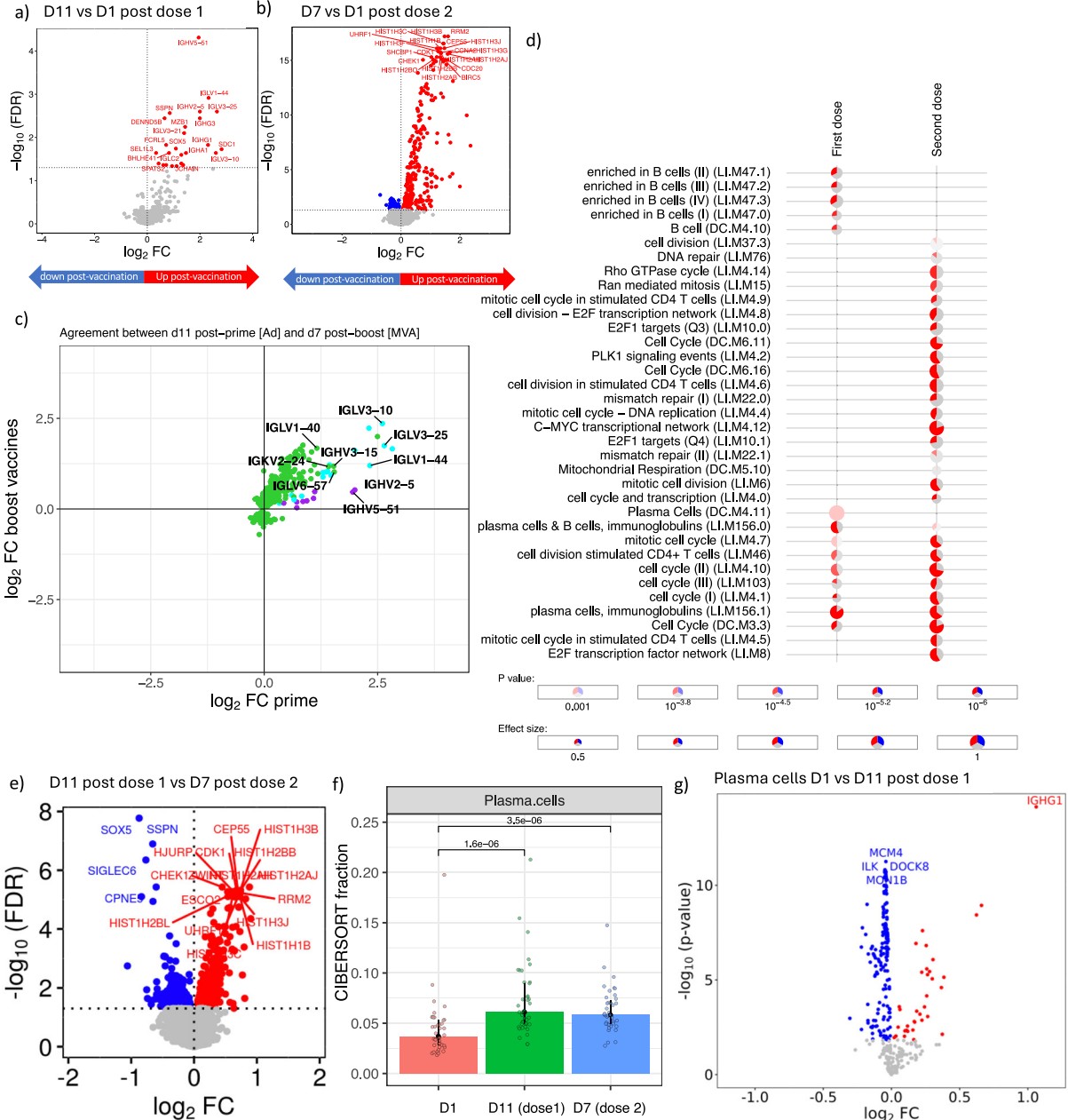

**Fig. 2 | The blood gene expression response to Ad26.ZEBOV and MVA-BN-Filo vaccines. a** Volcano plot highlighting differentially expressed genes (DEGs, false discovery rate [FDR] <0.05; red upregulated and blue downregulated) on day 11 after first dose of Ad26.ZEBOV vaccine (all groups) compared with pre-vaccination (25 DEGs, n = 40). **b** Volcano plot highlighting differentially expressed genes (DEGs, false discovery rate [FDR] <0.05; red upregulated and blue downregulated) 7 days after second dose vaccine (MVA-BN-Filo) compared with pre-vaccination (442 DEGs, n = 39). **c** Agreement plot of changes in gene expression (differentially expressed gene only) after the first and second dose of vaccine. In purple are DEGs after first vaccine, green represent DEGs after the second dose only and cyan are genes differentially expressed at both time points. **d** Modular signatures induced during different study time points, enriched modules (FDR < 1 × 10⁻³) are displayed order by q-value. Segments of the pie charts represent the proportion of upregulated (red) and downregulated (blue) genes (absolute fold change >1.25). Module

expression was assessed using the "tmodCERNOtest" function, this is a one-sided test for enriching that applies FDR correction for multiple testing. **e** Volcano plot highlighting differentially expressed genes (DEGs, false discovery rate [FDR] <0.05; red upregulated and blue downregulated) on day 11 after first dose of Ad26.ZEBOV vaccine (all groups) compared with 7 days after second dose vaccine (MVA-BN-Filo) (1987 DEGs, n = 39). **f** Plotted are the CIBERSORTx plasma cell fractions from whole blood RNA-sequencing data, with median and interquartile range. *P*-values were determined from a two-sample Wilcoxon rank-sum test (D1 n = 40 individuals, D11 n = 40 individuals and D7 n = 39 individuals). **g** Volcano plot derived from imputed plasma cell gene expression using CIBERSORTx on the 1000 most differentially expressed genes at day 11, highlighting differentially expressed genes based on two-sided moderated *t*-tests (DEGs, false discovery rate [FDR] <0.05; red upregulated and blue downregulated) day 11 after first dose of Ad26.ZEBOV vaccine (all groups) compared with pre-vaccination (228 DEGs, n = 40 individuals).

(Fig. 2c and Supplementary Fig. 2). Moreover, blood transcriptional module (BTM) analysis identified gene modules such as "plasma cells & immunoglobulins" and "cell division stimulated CD4 + T cells" that were upregulated following both study vaccines (Fig. 2d). Direct

comparison of post-first (+10 days) versus post-second vaccine (+7 days) also showed extensive differential regulation (1987 DEGs), albeit this appears to be mostly driven by the magnitude of changes (Fig. 2e and Supplementary Fig. 3). However, there was a notable

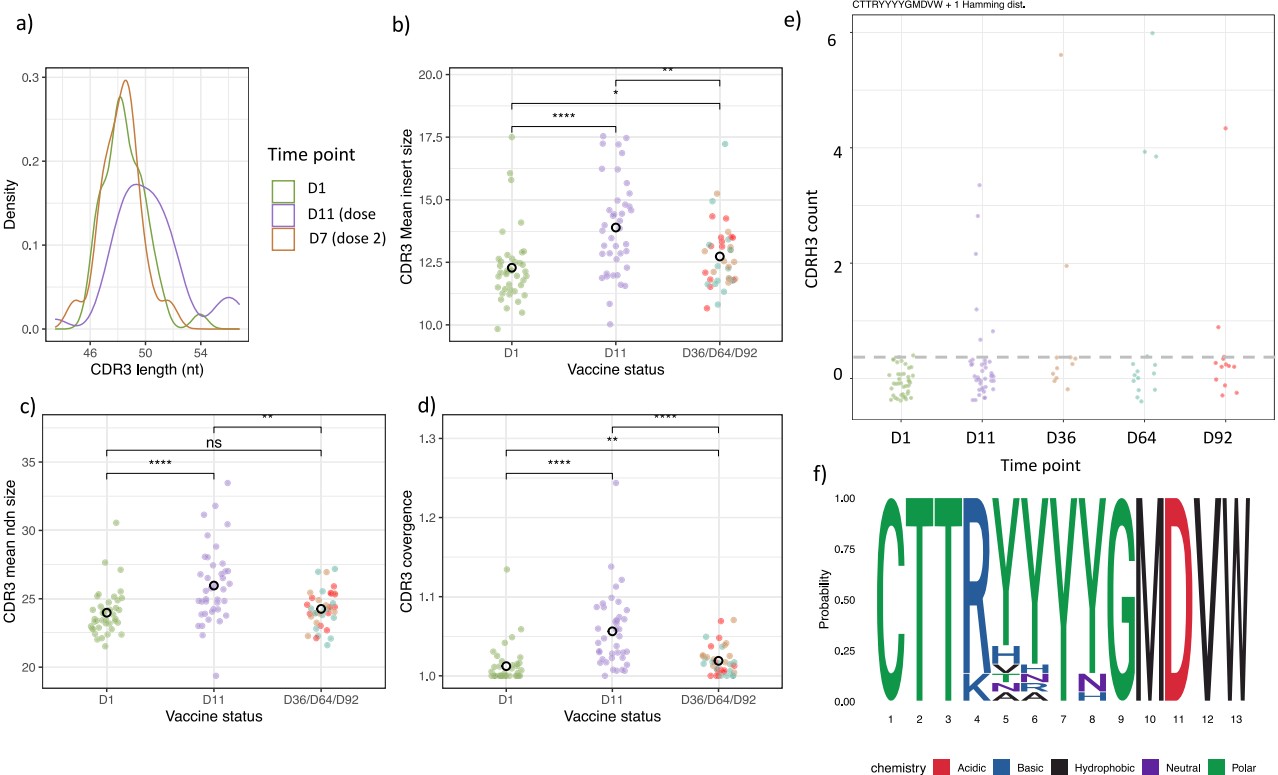

**Fig. 3 | The features of the B cell receptor sequence at the study time points.** **a** Plot shows the distribution of B cell receptor CDRH3 lengths at the study time points. **b** CDRH3 mean insert size at the study time points. **c** CDRH3 mean VJ junction (NDN) size at the study time points. **d** CDRH3 convergence at the study time points. **e** Identification of a CDRH3 sequence (and sequences within 1 hamming distance of this sequence) exclusively seen post-vaccination. Data points below the horizontal dashed line are 0. **f** The motif of the CDRH3 sequence (+1 hamming distance) of interest. A two-sided Wilcoxon rank-sum test was performed in (**b**–**d**): *: $p <= 0.05$, **: $p <= 0.01$, ***: $p <= 0.001$ and ****: $p <= 0.0001$.

difference in the expression of genes such as the transcription factor *SOX5* and several histones following each of the study vaccines (Fig. 2e and Supplementary Fig. 3). Also, a notable upregulation of cell cycle genes was observed after the second dose of vaccine (MVA-BN-Filo) (Fig. 2d and Supplementary Fig. 4). Cellular deconvolution analysis, using the whole blood transcriptomic data, indicated a relative increase in plasma cells both 10 days after the first vaccine and 7 days after the second vaccine (Fig. 2f). This analysis did not show consistent changes in other cellular fractions after both these vaccine doses (Supplementary Fig. 5). Moreover, extending the deconvolution analysis to impute cell type gene expression, implied that plasma cells were responsible for the upregulation in the immunoglobulin genes observed in these sequencing data (Fig. 2g and Supplementary Figs. 6 and 7).

**Detection of a complementarity-determining region 3 length (CDR3) exclusively seen post-vaccination with similar properties to GP-binders.** The upregulation of a shared set of genes encoding antibody segments following both vaccine doses prompted an investigation into whether this phenomenon reflected convergent, antigen-specific B cell receptor sequences. Given that immunoglobulin specificity is primarily determined by the hypervariable region (CDR3), we analysed CDR3 repertoire information extracted from the bulk RNA-sequencing data using the MiXCR software platform[13]. We extracted the B cell receptor sequences from the whole blood RNA-seq data and described study time point dependent differences in B cell repertoire properties: CDR3 length, insert size, VJ junction size and convergence (Fig. 3a–d). The CDRH3 length distribution shifted, and the VJ junction size increased 10 days post-vaccination compared with baseline (Fig. 3a, c). Additionally, both vaccine doses led

to an increase in the mean CDRH3 insert size and convergence relative to baseline (Fig. 4b, d). Moreover, a particular CDRH3 amino acid (+1 hamming distance) sequence was shared by several post-vaccination samples (13/79) that was not observed at baseline (0/40) (Fig. 3e). This CDRH3 used the IGHV3-15 variable gene, and the amino acid motif is shown in Fig. 3f.

**Machine learning model based on gene expression early post-vaccination able to predict subsequent antibody responses.** A machine learning model using the 200 genes most differentially expression 10 days after the first dose of vaccine was able to predict specific antibody responses (Fig. 4a, b). Analysis of the genes important in building this model included IGHV3-15 (ranked first in importance) and IGLV1-40 (ranked 11th in importance), these are the variable heavy and light chain genes used by an EBOV-GP binding monoclonal that is an edit distance of 2 from the CDRH3 uniquely seen post-vaccination in this data set (Figs. 3f and 4c). A machine learning model based on the most differentially expressed genes at 7 days post-second vaccine dose successfully predicted specific antibody responses to this dose (Fig. 4d, e). The genes with the highest importance in this predictive model are highlighted in Fig. 4f. Correlation coefficients between gene expression and subsequent antibody levels are shown in Supplementary Data 1 and 2.

**Single cell analysis reveals changes to plasma cell frequency, subclass usage and CDRH3 properties, following Ebola virus vaccines.** To further explore B cells responses to the heterologous Ebola virus vaccine regimen at high-resolution, we employed high-dimensional single-cell immune profiling using flow cytometric phenotyping complemented with single cell RNA-sequencing (scRNA-seq). PCA was

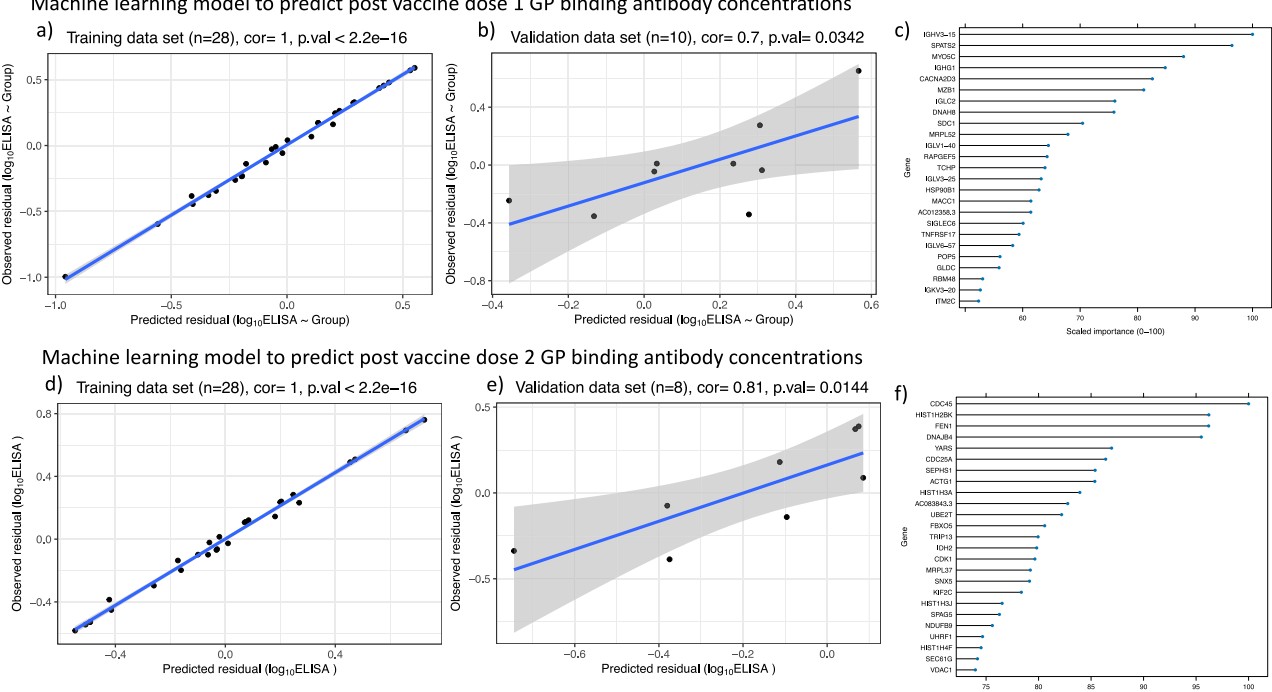

**Fig. 4 | Support vector regression (SVR) performance of model to predict post-vaccination EBOV glycoprotein binding antibody concentrations. a** training dataset built on the 200 most differentially expressed genes post-first dose of vaccine ($n = 28$), **b** performance of model in the test dataset post-first dose of vaccine ($n = 10$), **c** the top 25 genes ranked by importance from the SVR model predicting post-first dose EBOV glycoprotein binding antibodies. **d** training dataset built on the differentially expressed genes (FDR < 0.05) post-second dose of vaccine ($n = 28$), **e** performance of model in the test dataset post-second dose of vaccine ($n = 8$), **f** the top 25 genes ranked by importance from the SVR model predicting post-second dose EBOV glycoprotein binding antibodies. Two-sided Pearson correlation tests are used in (**a**, **b**, **d** and **e**) with the shaded area in shaded area representing the 95% confidence band for the predicted values from the linear model.

performed on the cytometric data (Total of 295,136 CD19⁺ viable B cells from 11 individuals), cell clusters were labelled as immature, naïve, unswitched memory, switch memory and plasma cells (Fig. 5a). Similarly, dimension reduction was performed on the scRNA-seq data from 34,344 B cells from 13 samples from 5 individuals collected on days 1, 11, and 64/92: cell clusters were labelled as immature, naïve, unswitched memory, switch memory, proliferating and plasma cells (Fig. 5b and Supplementary Data 3). The top gene markers that define scRNA-seq clusters via diferential expression—i.e., each cluster compared with all other cells—are shown in Fig. 6c. We did not observe consistent changes in the frequencies of total immature, naïve, switched memory or unswitched memory cells over the study time points by cytometry or scRNA-seq (Supplementary Fig. 8). The frequency of B cells expressing innate like B cell markers such as CD1d and CD11b also showed no overall change in frequency in the early days post dose 1 or dose 2 (Supplementary Fig. 20). However, increases in the relative frequency of plasma cells was observed for the majority of individuals 10 days after dose 1 and 7 days after dose 2 of the vaccination regimen—in both the cytometry and scRNA-seq data (Figs. 1f and 5d). In the scRNA-seq data, a nominally significant decrease in proliferating B cells (unadjusted $p$-value < 0.05) was also observed when comparing baseline samples with aggregated post-vaccination data (Fig. 5e). In the scRNA-seq data, differences in immunoglobulin subclass frequencies were evident between B cell clusters and study time point, with the proportion of IgG1 subclass being particularly enriched in plasma cells following the second dose of study vaccine (Supplementary Fig. 9). We also observed a statistically significant increase in the average CDRH3 length after the second dose of study for IgG, but not IgM, isotype plasma cells (Fig. 5f, g). The Hamming distance of the available CDRH3 sequences from each B cell population compared with a manually curated database of EBOV GP monoclonal antibodies is shown in Fig. 5h and

Supplementary Fig. 15. The minimal number of amino acid changes needed to match a known GP binding mAb was three, this was the case for two plasma cell CDRH3 sequences both seen post-vaccination (Fig. 5h—highlighted with blue arrow).

**Pseudobulk analysis reveals B cell subset specific vaccine-induced changes in gene expression including changes in export of protein from endoplasmic reticulum in plasma cells.** Principal component (PC) analysis shows separation of the plasma cell cluster on PC2—based on pseudobulk gene expression data (Fig. 6a). This plasma cell population was also distinct in terms of the cell cycle phase proportions, with enrichment of cells in G1 and relatively depletion of cells in S and G2/M phases (Supplementary Fig. 10). The largest number of DEGs post-vaccination in pseudobulk analysis were observed in the plasma cell population, with the majority of the DEGs seen 10 days after the first study vaccine also differentially expressed 7 days following the second study vaccine (Supplementary Fig. 13). Many of these shared DEGs encoded immunoglobulin segments (Fig. 6b, c). The biological processes inferred by gene set enrichment analysis of these plasma cells showed consistency following both vaccines, with 4/5 of the top upregulated gene pathways being shared (Fig. 6d, e). These pathways included "protein exit from endoplasmic reticulum", which is crucial for plasma cell production of antibodies (Supplementary Data 4 and 5).

**General agreement in gene regulation across cell types after study vaccines but disparities in immunoglobulin variable gene usage and cell cycle regulation.** The single cell RNA-seq data broadly indicated agreement in gene regulation, across the cell types, after each of the study vaccines (Supplementary Figs. 11 and 12). For example, in naïve and unswitched memory B cells, a common set of

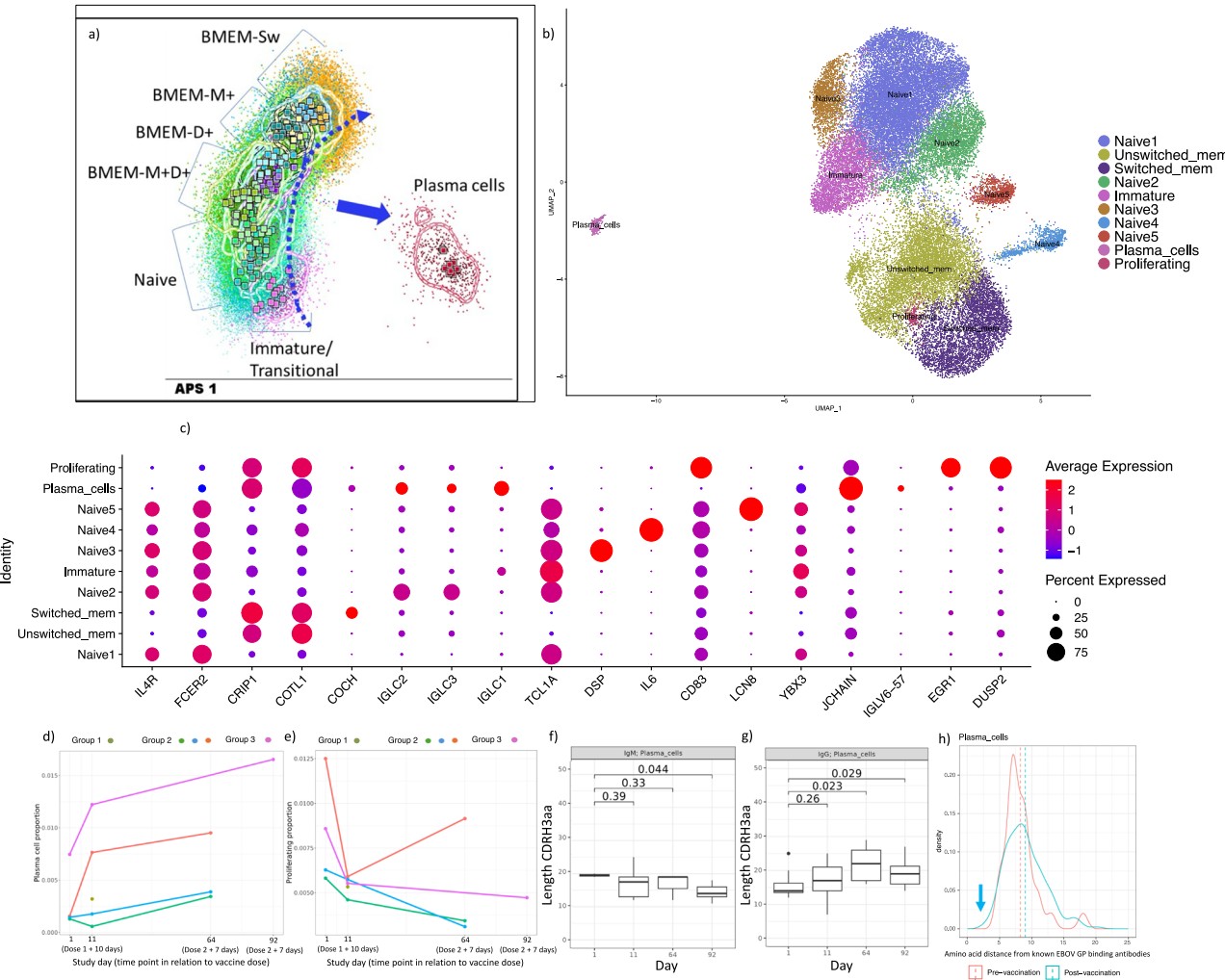

**Fig. 5 | The B cell characteristics by flow cytometry and single cell RNAseq.**
**a** APS-1 (automatic population separator) of the flow cytometric data based on the expression of surface markers listed in the methods, Table ii, (**a**) and Supplementary Fig. 19). The square symbols indicate each individual, coloured by population, and the contour lines indicate 1 × Standard Deviation for each population. The dotted arrow indicates the direction of differentiatin and the solid arrow the direction of plasma cell differentiation. **b** B cell clusters identified using 10× single cell RNA-sequencing (cells from all time points with immunoglobulin gene removed). **c** Top (n = 2) genes defining each B cell cluster (cells from all time points). **d** Plasma cell proportion of total B cells at study time points (each coloured line is an individual). **e** The proliferating B cell proportion of B cells at study time points (each coloured line is an individual). **f** CDRH3 length in IgM plasma cells at study time points (day 1 n = 2 cells, day 11 n = 6 cells, day 64 n = 3 cells, day 92 n = 9 cells). The horizontal lines are the median and interquartile range (IQR) with whiskers to +/− 1.5 × IQR. **g** CDRH3 length in IgG plasma cells at study time points (day 1 n = 8 cells, day 11 n = 37 cells, day 64 n = 7 cells, day 92 n = 32 cells). The horizontal lines are the median and interquartile range (IQR) with whiskers to +/− 1.5 × IQR. **h** Hamming distance of CDRH3 sequences from plasma cells to known Ebola glycoprotein binding antibody sequences. The two plasma cell CDRH3 sequences closest to known GP binding monoclonal antibodies (within 3 amino acid changes) are highlighted with a blue arrow. A two-sided Wilcoxon rank-sum test was performed in (**f** and **g**).

genes (*CDKN1A*, *LY9* and *PER1*) that were also differentially expressed after the first and second dose of vaccine (Supplementary Fig. 13a). *CDKN1A*—which encodes a protein involved in cell cycle G1 phase arrest—was downregulated post-vaccination and this coincided with an increase in the most abundant naïve B cell subsets in the S phase of cell cycle (Supplementary Fig. 10). However, differences in gene expression in the plasma cell population was seen both at the gene and pathway level, between the two study vaccines (Supplementary Figs. 11 and 14a). Several immunoglobulin gene segments were differentially expressed including paralogous IGHV genes, *IGHV3-53* and *IGHV3-66*, which were exclusively upregulated after MVA-BN-Filo (Supplementary Fig. 11). Moreover, disparities in the regulation of pathways related to cell cycle such as SCF SKP2 mediated degradation of P27/P21 were detected after the second dose of vaccine compared with the first vaccine, in naïve, unswitched and switched B cells (Supplementary Fig. 14b–d).

## Discussion

Here, we utilised contemporary approaches to dissect the B cells responses to heterologous two-dose schedule of Ad26.ZEBOV and MVA-BN-Filo Ebola vaccines. We describe an increase in total activated and migratory plasma cells following both doses of this vaccine regimen with an increase in the proportion of activated (ICOS+ PD1+) cTfh cells observed after the first dose of vaccine (Ad26.ZEBOV). An initial dose of Ad26.ZEBOV followed by MVA-BN-Filo induced a robust plasma cell response to the Ebola Zaire species glycoprotein. We revealed extensive differential gene regulation coinciding with the peak of plasma cell responses following these vaccines and were able to use machine-learning to interrogate gene expression data and predict subsequent antibody responses. Moreover, we detected a particular CDRH3 exclusively seen post-vaccination that had similar sequence properties to antibodies specific to Ebola virus GP. We went on use single cell approaches to describe changes to plasma cell

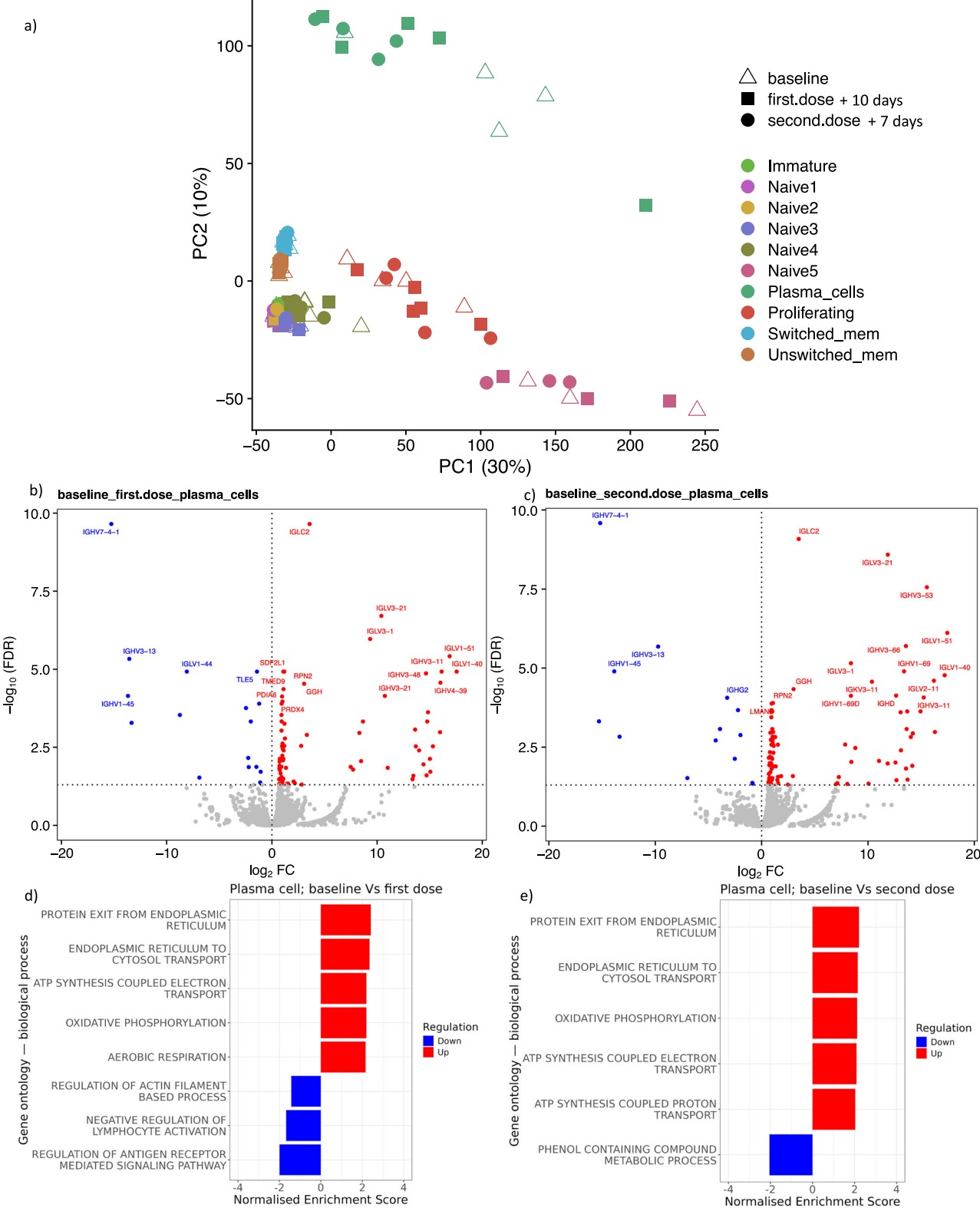

**Fig. 6 | Pseudobulk analysis of the B cell single cell RNA-sequencing data. a** First and second principal component (PC) of pseudobulk log10 counts per million expression values. **b** Volcano plot highlighting DEGs (false discovery rate [FDR] <0.05, red upregulated and blue downregulated) in plasma cells 10 days following the first study vaccine. **c** Volcano plot highlighting DEGs (FDR < 0.05) in plasma cells 7 days following the second study vaccine. **d** Gene set enrichment analysis on differentially expressed gene list from pseudobulk plasma cells 10 days following the first study vaccine compared with baseline, the most diffentially regulated upregulated and downregulated pathways are displayed (FDR < 0.05). **e** Gene set enrichment analysis on differentially expressed gene list from pseudobulk plasma cells 7 days following the second study vaccine compared with baseline, the most diffentially regulated upregulated and downregulated pathways are displayed (FDR < 0.05).

frequency, subclass usage and CDRH3 properties, following Ebola virus vaccines.

Our finding of robust Tfh responses is important because induction and maintenance of germinal centers is required for long-term B cell memory, long-lived antibody responses by bone marrow (BM) plasma cells and the boosting of antibody levels with subsequent immunisation or in response to infection[14–16]. Tfh cells and germinal centers have been shown to be required for memory B cells and long-lived antibody maintenance following the ChAdOx1 adenoviral vectored COVID-19 vaccine, a vaccine that induces robust BMEM responses and is associated with a slow rate of antibody decay[17–19]. SARS-CoV-2 vaccines based on the Ad26 backbone also induced memory B cell (up to 84 days post dose 1) and cTfh responses[20] as well as durable antibody titers[21,22]. We have shown here that Ad26.ZEBOV (dose 1) induces EBOV-GP specific BMEM, detectable in peripheral blood at d29 that increases further at d56 and still present at day 85. One hypothesis is that this is the consequence of prolonged antigen expression; in mice, non-replicating adenoviral vectors show persistent transgene expression at the vaccination site and in lymphatic tissues[23]. Evidence of prolonged germinal centers responses have been observed in mice following administration of human adenovirus expressing the model antigen OVA compared with OVA protein immunised mice[24].

We have also shown that Ad26.ZEBOV (dose 1) induces activated (ICOS⁺ PD1⁺) cTfh cells after immunisation (peaking 8 days after vaccination), but this was not evident up to 9 days following MVA-BN-Filo (dose 2). It has previously been described that adenoviral vectored vaccines induce strong primary humoral and cellular responses due to prolonged presentation of antigen[25,26]. This increase in activated cTfh cells post-first dose but not post-second dose is consistent with findings from other studies assessing these cells in an Ad followed by MVA (ChAd63-MVA) heterologous vaccination schedule[27]. In mice, MVA (MVA-OVA) appears to be a weak inducer of GC B cells[24]. Clinical studies have detected rises in cTfh cells when a priming MVA dose is given, but not after a repeated dose[28]. This may suggest that when MVA is given as a second dose it promotes differentiation of existing memory B cells into plasma cells rather than GC re-entry[29]. Nevertheless, we described boosting of BMEM after MVA-BN-Filo given as a second dose of vaccine, which is consistent with the literature of MVA-responsive IgG-secreting memory B cells following MVA (empty MVA) vaccine[28]. We also show that this long-lasting BMEM response was reproducible in participants from both UK and study sites in Uganda, Kenya and Burkina Faso. The memory B cell responses remained elevated above baseline at >4 years post dose 2 in the absence of further intervention. However, in contrast to the UK study, B cell memory responses were detected at lower frequencies in samples obtained after the first dose of vaccine in the African studies, despite the production of Ebola GP-specific antibody[7] which showed equivalent IgG GMC obtained in comparison to those see in the UK trial[6]. Despite the difference in Ebola GP specific memory B cell responses, we showed that total IgG-memory B cell responses were similar between all populations at all time points (Supplementary Fig. 16a). A direct comparison between study sites in the UK compared to the African sites was not made as the studies were not designed for this purpose. One explanation for this could be an influence of anti-vector immunity, as Ad26 neutralising antibodies (NAbs) are common in adult sub-Saharan Africans[30,31]. However, contradicting this explanation are data showing a second dose of adenovirus-vectored vaccines can boost B cell memory responses[32]. It is also possible that the smaller number of participants in EBL2002 vs EBL2001 (G1, n = 8 vs 13 and G2, n = 11 vs 13) combined with the data being only from the shorter dosing gaps contributed to lower median frequencies.

We described extensive differential gene regulation coinciding with the peak of plasma cell responses following Ad26.ZEBOV and MVA-BN-Filo vaccines. We observed general agreement in terms of the direction of gene regulation following both study vaccines, with the majority shared DEGs encoding immunoglobulin gene segments. However, direct comparison of gene regulation after each vaccine dose did highlight differences in a several histone genes as well as genes involved in cell cycle regulation. Moreover, the transcription factor SOX5 was upregulated following Ad26.ZEBOV compared with MVA-BN-Filo. SOX5 is involved in late B cell development and has a role in reducing proliferative capacity and is a marker of activated memory B cells[33,34]. Cellular deconvolution analysis indicated a relative increase in plasma cells following both vaccine doses, and that these cells accounted for the upregulation in the immunoglobulin genes. This is consistent with other studies of bulk whole blood sequencing data that show an increase immunoglobulin segments around a week after vaccination[35,36]. Moreover, we employed machine-learning to these changes in the gene expression post-vaccination to predict subsequent Ebola GP-specific antibody levels. Interestingly, IGHV3-15 and IGLV1-40 were ranked amongst the most important genes in this predictive model, and this variable heavy and light chain pairing is the most common combination in the publicly-available EBOV specific antibody sequences[37]. We extracted the B cell receptor sequences from the whole blood RNA-seq data. Post-vaccination, we observed a shift in the CDRH3 length distribution and an increase in both the mean CDRH3 insert size and VJ junction size, reflecting an adaptive response aimed at generating a broader repertoire of antigen-specific BCRs[38,39]. Additionally, the convergence of CDRH3 sequences post-vaccination suggests the selective expansion of vaccine-specific B cell clones. Moreover, we observed a cluster of highly similar CDRH3 sequences (within one hamming distance) that were present exclusively in post-vaccination samples. This CDRH3 sequence cluster contained the IGHV3-15 variable gene and its motif differed by only two amino acids, at positions 5 and 6, from a known Ebola virus glycoprotein binding antibody[37]. Central CDRH3 amino acid positions, such as positions 5 and 6, a tend to exhibit greater diversity, but their impact on CDRH3 structure and epitope interactions are not known[40]. However, CDRH3 sequences within this motif have been observed following EVD[41].

We next used single cell approaches to further explore B cells response to this heterologous Ebola virus vaccine regimen. A relative increase in the frequency of plasma cells in peripheral blood was observed for the majority of individuals 10 days after vaccine dose 1 and 7 days after vaccine dose 2 in both the cytometry and scRNA-seq data. This represents an increase in the total plasma cell frequency, coinciding with the peak in antigen-specific antibody-secreting cells in peripheral blood, which is not surprising as up to 90% of circulating IgG plasma cells 7 days post-vaccination may be vaccine-specific[42,43]. In the scRNA-seq there was also a decrease in "proliferating" B cells post-vaccination. This annotation was based on high expression of proliferation markers, such as MYC, and consistent with the description of phenotypically similar B cell population by ref. 44. On the UMAP clustering of scRNA-seq these cells reside between the unswitched and switched memory B cell populations. Although the function these cells has not yet to be described, expression of MYC by B cells has been shown to have a regulatory role in immunoglobulin class switch recombination[45]. Moreover, repression of MYC by BLIMP-1 is essential for differentiation of mature B cells into plasma cells[46]. Taken together, the increase in plasma cells and decrease in proliferating cells may represent the same underlying vaccine-induced differentiation of mature B cells into plasma cells.

Plasma cells 10 days after the first vaccine and 7 days after the second vaccine were enriched for IgG1 subclass usage, consistent with previously reports of an expansion of IgG1 plasma cells after other vaccines such as Tdap–IPV and BNT162b2[47,48]. The average CDRH3 length of IgG plasma cells increased after the second dose of vaccine, but not in IgM plasma cells. Changes in the distribution of

CDRH3 length can be indicative of an antigen-specific clonal expansion−as each clonotype would share the same CDRH3 length−which has been described following other vaccines[39]. A minimum of three amino acid changes is needed for one of these plasma cell CDRH3 sequences to match a known Ebola GP binding mAb−all of which were identified post-vaccination. Interestingly, both *VH3-53* and *VH3-66* gene segments were exclusively upregulated after MVA, these genes encode V regions that differ in only one amino acid position, in the framework region, and are functionally equivalent[49]. IGHV3-53 has previously been shown to be upregulated following MVA (MVA.HIVconsv), suggesting this observation may be vector specific[50]. Notably, in the scRNA-seq plasma cell data, we did not identify any CDRH3 sequences similar to the motif that was highlighted in the bulk RNA-seq data observed exclusively post-vaccination. However, the scRNA-seq experiment was limited by the scarcity of plasma cells among the total peripheral blood B cells with only 215 plasma cells being captured in the scRNA-seq experiment. This low number may particularly limit the generalisability of the pseudobulk plasma cell analysis, as the comparison was based on a small number of cells. In addition, the scRNA-seq was run on a smaller number of individuals ($n = 5$) than the bulk RNA-seq experiment ($n = 42$). The modest sample size, coupled with a high number of parameters, likely contributed to some degree of overfitting in the machine learning model, as evidenced by higher predictive accuracy in the training set compared to the test set. Nevertheless, the model was able to explain over 50–60% of the variance in post-vaccination ELISA levels, and the predicted values remained significantly correlated with the observed values. These findings suggest that, despite the potential for overfitting, the model retains substantial predictive value and offers meaningful biological insights.

Further limitations to this study include that our transcriptomic and cellular phenotyping assessment of B cells following the two doses of study vaccines was not antigen-specific. However, previous studies that have suggested up to 90% of IgG-secreting antibody-secreting cells 7 days after vaccination are antigen-specific[43]. Accordingly, we were able to demonstrate changes in both the frequency and properties of plasma cells in blood 7 days post-vaccination. A limitation to our comparison of immune responses to the first and second dose of study vaccine is that these vaccines differed both in terms of viral vector and transgene antigens. The first adenovirus vectored vaccine contained only one Ebola virus GP, whereas the second MVA vectored vaccine contained two additional GPs and a nucleoprotein from other Ebola virus variants. Prior to the first dose participants will have been immunologically naïve to both the viral vectors and transgene; however, the immune response to the second vaccine will be a conflated de novo and recall immune response. Nevertheless, this heterologous and multivalent vaccination strategy produces robust humoral and cellular responses to multiple *Orthroebolavirus* species[6,51]. Homologous 2-dose regimens (e.g., Ad26, Ad26 or MVA, MVA) are less immunogenic than heterologous strategies but induce humoral immune memory[52]. Since homologous 2-dose regimens have not been examined at the resolution of B cell subset-specific responses (e.g., plasma cells, memory B cells) or through RNA-seq analysis, direct comparison with the data presented here is not possible.

EBOV-GP is a glycosylated viral surface protein which contains a fusion domain, that exists in both membrane-bound and cleaved (secreted) forms, the latter being present during infection, and both versions can bind to TLR4 via the GP-fusion domain[53]. TLR4 is expressed by dendritic cells and macrophages[54], as well as T cells, B cells and NK cells[55,56]. However, in vitro studies involving direct stimulation of T, B and NK cells with purified, soluble glycoprotein exhibited increased cell death[56,57]. In human B cells, TLR4 expression is induced following activation[58]. In mice, the combination of antigen plus TLR7 and TLR4 ligands results in increased specific, NAbs and long-lasting germinal center responses and similar effects were seen in humans, involving TLRs on B cells, DCs and activation of T cells[59]. It is possible that prolonged presentation of antigen to activated B cells and T cells expressing TLR4 has contributed to the persistent circulation of high frequencies of BMEM seen in this study. Therefore, a proposed mechanism for long term maintenance of elevated BMEM frequencies involves the adenoviral vectors infecting DCs, resulting in DC expression of EBOV-GP, providing the initial combination of antigen plus multiple TLR stimuli and activated T cell cytokine production to initiate naïve B cell activation. Once this has been initiated the BMEM derived from this priming event upregulate TLR4 and migrate to other lymphoid tissues[60]. The prolonged expression of antigen in the resulting germinal centers[24] may continue to stimulate this response in the subsequent days prior to the second dose of vaccine (MVA-BN-Filo).

Similar requirement for antigen-specific stimulation of murine B cells via TLR4, in combination with T cell production of cytokines such as IL-2 was also shown to drive plasma cell production[61]. In a murine immunisation study specifically targeting MAdCAM-1+ stromal cell TLR4 with adjuvants induced germinal center responses with increased GC B cells, BM long-lived plasma cells and enhanced antibody production in both young and aged mice[62]. We demonstrated further evidence of Ad26.ZEBOV priming a germinal center response with the appearance, in peripheral blood, of IgG secreting plasma cells (IgG-ASC), mirrored by the by the appearance of CD19 + CD20loCD27 + CD38 + CD24− plasma cells expressing CD86, α4β1 or CD62L. Transient increase in migratory, vaccine-specific plasma cells can be used to detect the presence of ongoing vaccine responses with surface markers such as CD62L, indicative of antigen-specific migratory plasma cells[63], CD86, highly expressed on recently activated plasma cells[64] and α4β1 (VLA-4), which enhances bone marrow survival[65]. This response may be related to the viral vector as the EBOV-GP specific-IgG-ASC was weak. Previously, it has been shown that cytopathic viruses such as VSV and influenza promote both extrafollicular and germinal center responses, resulting in rapid neutralising antibody responses and GC responses lasting for up to 100 days[66]. The weak EBOV-GP-IgG-ASC response, which we show in this study (at day 9–13), may represent the extrafollicular response to dose 1. The more robust responses after MVA as dose 2, peaking at day 7, resulted from the prolonged germinal center responses following dose 1, deriving from the continued generation of BMEM. Previous description of blood plasma cell responses to toxoid vaccines (tetanus/diphtheria) described the CD62L+ Plasma cells as being enriched for IgG+ antigen-specific cells[63]. Also the enhanced expression of CD86 on plasma cells inparticular has been shown to be essential for IgG secretion[64,67,68]. While expression of α4β1 has been shown to upregulate in response to CXCL12 and has been implicated as a plasma cell survival molecule, interacting with VCAM-1 on bone marrow stromal cells, thus suggesting that the plasma cells characterised here represent the bone marrow[65] homing, antigen-specific plasma cells which were detected in the plasma cell ELISPot. Data from mice immunised with AdHu5 showed that viral vectors induced strong germinal center B cell and Tfh responses, with antigen-specific cells detected by days 7–11 post immunisation and antigen-specific germinal center B cells still elevated by day 34 when the experiment ended[24].

Here, we demonstrate that a heterologous EVD vaccine regimen (Ad26.ZEBOV, MVA-BN-Filo) induces robust plasma cell responses as well as prolonged B cell memory. Moreover, we revealed early immune responses, in peripheral blood, that were vaccine-specific and predictive of the magnitude of subsequent antibody responses. These insights into the induction of strong and persistent vaccine-induced B cell immunity that will inform future preparedness strategies against diseases with outbreak potential, such as EVD.

## Methods

### Participants, vaccine schedule and blood Sampling

Blood samples were collected from healthy, UK, adult participants (aged 18–65) as part of the VAC52150 Ad26.ZEBOV, MVA-BN-Filo, randomised, observer blind, placebo-controlled, Phase 2 study (ClinicalTrials.gov NCT02416453, EudractCT 2015-000596, which was undertaken as part of the Innovative Medicines Initiative-2 EBOVAC2 consortium European for evaluation of prophylactic vaccines for EVD. The study was approved by UK National Research Ethics Service (South Central, Oxford; A 15/SC/0211). We have obtained informed consent from all participants. Participants were enrolled as part of a multi-center trial at University of Oxford (UOXF), (UCL) and IMSERM, France and the clinical trial is reported[6]. The participant samples studied in this report were all recruited at UOXF and were divided into two study Cohorts (Supplementary Fig. 1).

A subset of participants (n = 18, six per group) were recruited to a follow-up study PRISM, at approximately 4 years (41 months) post dose 2, to look at maintenance of the immune response in UK Adults. Two blood samples were obtained, V1 (41 [39–43] months post dose 1, and V2 (201[181-239] days post V1. Memory B cell IgG responses were quantified using the described ELISpot assay (Supplementary Fig. 1). The memory B cell responses were also analysed from a subset of samples collected from healthy adults, (aged 18–70 years old) as part of a Phase II Trial of the same vaccines, which was conducted at multiple sites across Africa (Kenya, Uganda and Burkina Faso) under EBL2002, ClinicalTrials.gov NCT02564523[7]. The samples were obtained from the Janssen central biobank CSM Europe, under CUREC R73519/RE001 (Supplementary Fig. 1).

### Peripheral blood mononuclear cell (PBMC) isolation

PBMCs were isolated and frozen within 5 h of venepuncture. Whole blood was centrifuging at $1800 \times g$ for 10 min, removing plasma for freezing. Resuspending cells in Rinse buffer (Miltenyi, UK) + heparin, pouring into leucosep tubes (Greiner), centrifuging at $1000 \times g$ for 20 min (break off). Remove PBMC layer, resuspend in rinse buffer, wash at $250–300 \times g$ for 10 min, remove supernatant. Incubate with RCLB (Qiagen) for max 5 min, resuspend to 50 ml with rinse buffer. Wash at $250–300 \times g$ for 10 min. Resuspend to count with Millipore Sceptre Cell Counter. Pellet cells, remove supernatant and resuspend in Recovery™ Cell Freezing Medium (Gibco) at $10 \times 10^6$ cells per/ml. Frozen down at −80 °C in Mr Frosty overnight and transferred to LN2 within 72 h.

### ELISpot Assay for detection of EBOV-GP specific IgG-ASC

ELISpot plates (MSCRN-IP Dura 0.45UM, 96-well PVDF membrane, Merck-Life Technologies #MAIPS4510) were coated with ZEBOV-his-GP (5 μg/ml, donated by Janssen), Anti-Ig (10 μg/ml), Tetanus Toxoid (5 μg/ml) and PBS only. The plates were coated for at least 24 h prior to use and kept at 4 °C for 1 week. Prior to use, the plates were washed three times with sterile PBS (200 μl/well) and then blocked for at least 1 h with CM.

For ex vivo plasma cell detection on Cohort 1 and 2, freshly isolated PBMCs (d1, d9, 11, 13 and 15 post dose 1 and d1, 3, 5, 7 and 9 post dose 2). PBMCs were resuspended at $2 \times 106$ cells/ml CM and the $2 \times 10^5$ cells added to ZEBOV (triplicate), Tetanus Toxoid (duplicate) coated wells. For Total IgG binding $2 \times 10^6$ cells were diluted at 1/10, 1/100. The plates were then incubated overnight (16 h) at 37 C/5%CO2/95% humidity.

Polyclonal stimulation of PBMCs to expand memory B cells to detect ZEBOV specific IgG-antibody-secreting cells (ASC) by ELISpot. PBMCs (following defrosting and overnight rest) were added to plates at $2 \times 10^5$ cells/well in CM. Then 100 μl of polyclonal stimulation mix was added per well, final concentrations [Staphylococcus aureus Cowan Strain (SAC) 1:5000, Pokeweed Mitogen 83 ng/ml and CpG-ODN-2006 (1.7 μg/ml)] and then incubated for 5 days at 37 C/5%CO2/

95% humidity. Following the incubation period cells were pooled and washed two times in running buffer at $300 \times g$ for 10 min, counted and resuspended at $2 \times 10^6$ viable cells/ml CM and 100 μl added per well for ZEBOV (quadruplicate), tetanus toxoid (duplicate), PBS (duplicate) and for total IgG binding at 1/100 and 1/1000 in duplicate. The plates were incubated overnight.

Developing ELISpot plates for plasma cell and memory B cell assay: Cells were flicked out of the plated, tapped on tissue and then washed four time with PBS + 0.05% Tween, and once with PBS. Goat-Anti-human IgG-Alkaline Phosphatase conjugate was diluted 1:5000 in CM, filtered and 50 ul/well added to all test wells. The conjugate was incubated for four hours at room temperature, followed by washing as above followed by one wash with water. Biorad Alkaline substrate was made up immediately before use in room temperature water, filtered and 50 μl added per well. Development of the substrate was monitored and stopped with 200 μl/well water when spots appeared, before background turned purple. The plate was then washed three time with water and dried overnight prior to reading.

Plates were analysed using AID ELISpot reader and software with pre-set count settings. Debris and background was removed prior to data export in excel and analysis and statistics in GraphPad PRISM V10.

### Whole blood bulk RNA-sequencing

Venous peripheral blood (up to 3 ml) was collected directly in Tempus™ RNA tube immediately pre-vaccination, then 10 days post-first vaccine and 7 days post-second vaccine. RNA was extracted using the Tempus™ spin RNA isolation kit (Invitrogen, USA). Eluted RNA was quantified using Qubit™ RNA BR Assay Kit (Invitrogen, USA) and quality control was conducted using the Agilent RNA ScreenTape assay on the 4200 TapeStation System (Agilent, USA). RNA was ribodepleted and globin depleted using Ribo-Zero™ Gold rRNA removal Kit (Illumina, USA). RNA was converted to cDNA, second-strand cDNA synthesis incorporated dUTP. The cDNA was end-repaired, A-tailed and adapter-ligated and prior to amplification, samples underwent uridine digestion. The prepared libraries were size selected, multiplexed and quality controlled before 75 bp paired-end sequencing (HiSeq4000). Sequencing was conducted at the Wellcome Trust Centre for Human Genetics (Oxford, UK). The sequencing data were aligned against the whole human (*Homo sapiens*) genome build GRCh38 (https://ccb.jhu.edu/software/hisat2/index.shtml), using STAR (version 2.6.1d). Gene features were counted using HTSeq (version 0.11.1), using human gene annotation general transfer format version GRCh38.92 (www.ensembl.org). Genes with low counts across most libraries were removed by only retaining genes with abundance greater than 3 counts per million in 4 or more samples. Ribosomal RNA (rRNA), sex chromosome genes, mitochondrial RNA and haemoglobin genes were excluded from downstream analysis. Human leukocyte antigen typing of RNA-sequencing data was used to check correct pairing of samples collected from the same participants using RNA2HLA (version 1)[69]. The analysis script is deposited on GitHub (https://github.com/dan-scholar/Ebola_vaccine_B_cell.git).

### Statistics & reproducibility

**Gene expression analysis.** Differential gene expression analysis was performed using the R Bioconductor packages edgeR and limma[70–72]. RNA-sequencing data were normalized for RNA composition using the trimmed mean of *M*-value method[73]. The data were then transformed using the voom function in limma. A linear model was fitted to the data with the lmFit function, employing the empirical Bayes method to share information across genes[74]. Paired analyses were conducted to compare pre- and post-vaccination samples at each study time point, with a statistical significance threshold set at a false discovery rate (FDR) < 0.05.

**Blood transcriptional module analysis.** BTM analysis was performed using the "tmod" R package, with genes ranked based on their log-ratio (LR) values. Module expression was assessed using the "tmodCER-NOtest" function, a non-parametric statistical test that operates on gene ranks[75].

**Machine learning derived predictors of immune responses.** We developed a supervised machine learning algorithm to predict GP-specific antibody levels based on changes in differentially expressed genes post-vaccination. A linear kernel support vector regression (SVR) model was trained on 75% of the data, using fivefold cross-validation repeated 10 times to optimize performance. The trained model's performance was then evaluated on the remaining 25% of the dataset. A support vector regression (SVR) model was developed to predict antibody levels following the first vaccine dose. The dependent variable comprised the residuals of log10-transformed ELISA measurements, with the group effect regressed out. Model features included post-vaccination limma voom-normalised expression ($E$) values from the top 200 differentially expressed genes (DEGs) identified after the first dose. A similar modelling approach was applied to predict antibody levels following the second vaccine dose, using all DEGs identified post-second dose as input features. A linear SVR was selected due to its robustness in high-dimensional settings, where the number of features exceeds the number of observations. In such contexts, linear models are less prone to overfitting and often perform more reliably. Moreover, SVR with a linear kernel offers greater interpretability compared with many non-linear alternatives, allowing clearer insight into the contribution of individual features to the predicted outcome. The full analysis script is deposited on GitHub (https://github.com/dan-scholar/Ebola_vaccine_B_cell.git).

**Cell abundance deconvolution from whole blood transcripto mic data.** The cell composition of whole blood samples was assessed using the CIBERSORT× method[76]. A filtered, non-log-transformed reads per kilobase million (RPKM) sample gene matrix, along with a "signature" gene file (LM22; representing 22 immune cell types), was used to deconvolute cell abundances. CIBERSORTx was executed in relative mode with B-mode batch correction and 1000 permutations.

**B cell receptor profiling from bulk RNA-sequencing data.** B cell receptor sequence information was extracted from bulk RNA-sequencing using the MiXCR (MIXCR (version 2.1.12) functions "Align," "Assemble partial," "Assemble," and "Export clones" using default settings[13]. Basic statistics were calculated on MiXCR output using VDJtools (version 1.1.10)[77].

**Single cell B cell RNA-sequencing.** B cells were enriched from cryopreserved peripheral blood mononuclear cells. PBMCs were labelled with CD19-BB515, and [CD3 + CD4 + CD14 + CD16 + BV421 and viability stain] as a dump channel. Viable, CD19 + B cells were sorted using FACS Aria III with 85 μM sort nozzle, FACS Diva software. B cells were sorted into complete medium (RPMI + 10%FBS with pen/strep, non-essential amino acids, sodium pyruvate and 2-mercaptoethanol) and transferred directly to the sequencing facility for 10× sequencing.

**Data preprocessing.** Single cell libraries were created using a Chromium 10× controller using 5'gene expression (v2.0) kits and Human V(D)J B cell (v2.0 kits). Sequencing was performed using 150 paired-end sequencing on the NovaSeq 6000 system (Illumina, USA). Single-cell RNA-sequencing data was pre-processed using CellRanger (10× Genomics, version 6.1.1) and aligned to the GRCh38 genome. Clonotype group was inferred using Enclone, CDR3 nucleotide identity of at least 85% was required for subclonotype retention (10× Genomics, CellRanger version 6.1.1).

Single cell RNA-seq data analysed using the Seurat package (version 4.3.0) in R (version 4.3.1). Quality control was performed; cells with between 300 and 4000 detected genes (nFeature_RNA) and <5% mitochondrial genes were concerved. Log normalisation was performed on gene counts for each cell, divided by the total counts for that cell and then multiplied by 10,000 (scaling factor). Features were scaled and centerd using the ScaleData function. The function ElbowPlot was used to visualise the dimensionality of the data and select the number of PCs to use in subsequent unsupervised analyses. Batch correction was performed using Harmony (version 0.1.1). As previous reports have shown immunoglobulin genes interfere with biologically meaningful unsupervised clustering[78], these genes were removed for the purpose of cell cluster identification (using function! grepl("IG[HKL][VDJ]|IGHM|IGHD|IGHE|IGHA[1,2]|IGHG[1–4]|IGKC| AC233755.1"). Cell clusters were defined using FindClusters function using a shared nearest neighbour graph constructed on the harmony batch corrected reduction generated using the FindNeighbors function. B cells were selected based on being *MS4A1* positive as well as being *GNLY*, *CD3E*, *CD14*, *FCGR3A*, LYZ, *CD8A*, and *PPBP* negative. As plasma cells express a low level of *MS4A1*, a population of *MS4A1* low/ negative but *JCHAIN* positive/high cells were also selected (i.e., negative for the aforementioned non-B cell markers). Batch correction and cell clustering was repeated (as previously described) to identify B cell subpopulations. The V(D)J information was incorporated from the B cell receptor (BCR) sequencing libraries and all B cell populations were checked to have >90% of cells with a successful BCR clonotype identified.

**B cell clustering.** B cells were manually categorised into six sub-populations: immature (*IGHD* + *CD27*−*PPP1R14A*+), naïve (*IGHD* + *CD27*−*PPP1R14A*−), unswitched memory (*IGHD* + *IGHM* + *CD27*+), switched memory (IGHD−*CD27* + ), plasma cells (*MS4A1*$^{low}$ *XBP1*+) and proliferating B cells (*IGHD*$^{low}$*IGHM* + *CD27*+ *MYC*$^{high}$). Subpopulation cluster markers were identified by comparing gene expression in each cluster with all remaining cell using the FindAllMarkers function. The analysis script is deposited on GitHub (https://github.com/dan-scholar/Ebola_vaccine_B_cell.git).

**Cell cycle stage.** It is possible to infer the cell cycle from scRNAseq data based on the expression of G2/M and S phase markers, with those cells neither of these being designated as G1. Cell cycle scoring was completed using Seurat::CellCycleScoring.

**Reporting summary**

Further information on research design is available in the Nature Portfolio Reporting Summary linked to this article.

## Data availability

The gene expression data generated in this study have been deposited in the Gene Expression Omnibus database under accession code (GSE273114, https://www.ncbi.nlm.nih.gov/geo/query/acc.cgi?acc=GS E273114). Source data are provided with this paper.

## Code availability

The reproducible code for analysis is available on GitHub (https://github.com/dan-scholar/Ebola_vaccine_B_cell.git).

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

## Acknowledgements

The study was coordinated by the EBOVAC2 multi-partner research consortium, which received funding from the Innovative Medicines Initiative 2 Joint Undertaking (grant number, 115861) as part of the IMI Ebola+ Programme. This Joint Undertaking receives support from the EU's Horizon 2020 Framework Programme for Research and Innovation and the European Federation of Pharmaceutical Industries and Associations. We are grateful to the volunteers who participated in this study. We thank Theodosios Kyriakou and the Oxford Genomics Centre at the

Wellcome Centre for Human Genetics (funded by Wellcome Trust grant reference 203141/Z/16/Z) for the generation and initial processing of the sequencing data. This research was funded in part by the National Institute for Health Research (NIHR) Oxford Biomedical Research Centre. The views expressed are those of the author(s) and not necessarily those of the NHS, the NIHR or the Department of Health.

## Author contributions

D.O.C., E.A.C. and A.J.P. conceived and designed this work. E.A.C., S.B., K.A.S., and R.M. performed the laboratory experiments described. D.O.C., E.A.C., D.F.K. and A.J.P. made substantial contributions to interpretation of data. D.O.C. and E.A.C., wrote the initial draft of the manuscript—M.M.G., S.B., K.A.S., R.M., D.F.K. and A.J.P. substantively revised it. All the authors reviewed and approved the final version of the manuscript. All authors approved the submitted version of this manuscript and have agreed both to be personally accountable for the author's own contributions and the integrity of the work.

## Competing interests

A.J.P. is chair of the UK Department of Health and Social Care's Joint Committee on Vaccination and Immunisation. A.J.P., S.B., E.A.C., K.A.S., R.M. and D.O.C. are contributors to intellectual property licensed by Oxford University Innovation to AstraZeneca. The remaining authors declare no competing interests.
