## [Transparent Peer Review file · Nature Communications]

Prediction and characterisation of the human B cell response to a heterologous two-dose Ebola vaccine

Corresponding Author: Dr Daniel O'Connor

Version 0:

Reviewer comments:

Reviewer #1

(Remarks to the Author)

Summary: The manuscript Phenotypic and transcriptomic characterization of the human B cell response to a novel Ebola vaccine regimen (Ad26.ZEBOV, MVA-BN-Filo) by O'Connor et al. utilized cellular, transcriptomic, and machine learning to fully characterize the B cell phenotypes stimulated and associated follicular T cell populations that contribute to the B cell population formation, following heterologous Ad26.ZEBOV and MVA-BN-Filo vaccination in two clinical cohorts. The authors were able to identify a transcriptomic signature that correlated with magnitude of antibody responses. Deeper analysis into the BCR repertoire revealed a unique CDRH3 sequence that is similar to previously described antibody sequences. The authors utilized this unique sequence to further characterize the B cell cellular populations associated with this unique CDRH3 sequence post vaccination. This study not only describes the unique B cell repertoire, but it also compares the similarities of the stimulated immune response with clinical cohorts in Europe and Africa to ensure the validity of the described results in the effected country. Overall, a very through transcriptomic approach to B cell characterization in multiple human cohorts demonstrating strong associations across cohorts to ensure predictive power of their model system moving forward. My suggestion is for minor revisions before being accepted for publication.

Major comments:

1. Is there any way for the authors to include archived data from a homologous vaccination clinical cohort to delineate differences in the heterologous vaccination strategy?
2. In the unique CDHR3 sequencing key amino acids 5 and 6 have multiple sub variations possible can the authors comment on the impact of the less common amino acid sequences and how they could potential contribute.
3. Can the authors use the BCR sequencing as an additional filter to determine antigen specificity during the analysis?

Minor Comments:

- 1.) Zaire ebolavirus is old nomenclature adjust to current nomenclature.
- 2.) At the end of the introduction emphasize the samples are from a human clinical cohort don't downplay the importance of your samples.
- 3.) Result section one: change follow-on study to follow-up study.
- 4.) Figure 2 and Supplemental font different.
- 5.) Remove the syringe art in figure 1.
- 6.) Supplemental 16-18 reference font different.
- 7.) Keep day abbreviation vs writing it out consistent throughout the manuscript to goes back and forth.
- 8.) Discussion correct centres to centers and titres to titers.

Reviewer #2

(Remarks to the Author)

BRIEF SUMMARY:

O'Connor / Clutterbuck et al. have investigated the characteristics of Ad26.ZEBOV, MVA-BN-Filo vaccine immune responsiveness from UK/Africa clinical trial samples in a way that attempts to go into more depth than other previous publications using these same clinical trial samples. They assess gene expression and abundance changes of B cells, T follicular helper cells and plasma cells over time with FACS and EliSpot after the first and second vaccine dose. They

determine differentially expressed genes of participants after the first and second dose with bulk RNA-seq. They also use bulk RNA-seq alignments to characterize changes, after the first and second dose, in B cell receptor CDRH3 length and sequence, finding a motif after vaccination that is similar one that has been implicated in Ebola virus glycoprotein binding. Furthermore, they developed an algorithm to predict vaccine immune responsiveness, using machine learning that is trained on the RNA-sequencing data to find the top genes that best predict this responsiveness. Finally, they used single-cell analyses with FACS cell-type enrichment and PCA/UMAP with single cell RNA-sequencing, assessing gene expression of different cell types and changes in cell-type/subtype abundance and gene expression after vaccination. Understanding an immune system's response to vaccination is incredibly important, and particularly for immunization against the Ebola virus, which is unfortunately too common in some African countries, highly contagious, and deadly.

OVERALL ASSESSMENT:

While the topic of this manuscript is highly important, the content is not distinctively novel or cohesive, and it lacks comprehensiveness. The manuscript may not yet meet the standard expected for a publication in Nature Communications. Significant revision is needed, including more detailed descriptions, enhanced detail, and improved organization. I hope the following comments will be of help to improve the manuscript.

MAJOR COMMENTS:

1.
Lines 79-90: Should mention the staggered timing of the 2nd dose in the manuscript text and a brief rationale for testing this variable.
2.
Lines 82-88. You describe Cohorts 1 and 2 but not Cohort 3. Are "cohorts" the same thing as "groups"?
3.
Lines 236-242: Observations/conclusions for Figure 4a, 4b, 4c, 4d should be stated in the manuscript.
4.
Missing from methods:
Differential expression analyses – how? what script/package/parameters?
blood transcriptional module (BTM) analysis – how? what script/package/parameters?
CIBERSORTx – how was it use? parameters?
CDR3 length, insert size, ndn size, convergence, motif analyses – how were these analyses done after the Mixcr output of the receptor sequence?
5.
Line 666: Nothing is written under "Machine learning derived predictors of immune responses". How was this done? Did you build it off of a previous study? Explain your methodology in detail.

MINOR COMMENTS (AND SUGGESTIONS):

6.
Introduction - lack of citation in the introduction for clinical trials and other publications, eg.
[https://doi.org/10.1016/S1473-3099\(24\)00058-6](https://doi.org/10.1016/S1473-3099(24)00058-6)
<https://doi.org/10.1371/journal.pmed.1003865>
<https://doi.org/10.3389/fimmu.2023.1215302>
<https://doi.org/10.1080/21645515.2017.1264755>
7.
Figure 1:
Can get quite messy when the staggered timings and the "common" timings are all put together. Could be helpful to visually distinguish them (maybe a box around the "common" timings).
8.
Figure 2:
For easier read, could be helpful to write dose 1 and dose 2 under the figure panels e-g to distinguish, or specify dose 1 (left), dose 2 (right) in legend
f) Helpful to write more precisely in legend what surface markers mean, e.g., subtype CD86 (activated), subtype CD62L (ag specific migratory), and a4β1pos (enhanced survivability)
9.
Line 134: Mistake in Figure 2 legend
e) Ex vivo ELISpot Frequency of Total IgG-ASC and g) EBOV-GP specific IgG-ASC should be
e) Ex vivo ELISpot Frequency of Total IgG-ASC (top) and EBOV-GP specific IgG-ASC (bottom)
10.
Line 158: 2f cited before 2e in the paper – "Figures and tables in the main manuscript must be cited in the order they appear

in the text, figure legends, table legends and boxes”

11.

Line 176-180: Transient increase in migratory, vaccine specific plasma cells can be used to detect the presence of ongoing vaccine responses with surface markers such as CD62L, which is indicative of antigen specific migratory plasma cells (Mei et al., 2009), CD86, which is highly expressed on recently activated plasma cells (Rudolf-Oliveira et al., 2018) and $\alpha 4\beta 1$ (VLA-4) which enhances bone marrow survival (Khodadadi et al., 2019).

12.

Line 189: Add: While these findings are not statistically significant, they do point to a trend of plasma cell responses to both dose 1 and dose 2 administration.

13.

Line 193: Change “We next explored gene regulation in whole blood samples” to “We next explored gene regulation with bulk RNA sequencing of whole blood samples”

14.

Line 197: sixteen genes not “sixteen gene”

15.

Figure 3:

a) DEG dose 1 –change title to “D1 vs D11 post dose 1”

b) DEG dose 2 – change title to “D1 vs D7 post dose 2”

c) Colors for points feel arbitrary– could have (a) title be one color, (b) title be another color, then these refer to the points in c

d) Organization of modular signatures – is it from most to least enriched? Alphabetical? Largest to smallest group of genes?

e) Title “D11 post dose 1 vs D7 post dose 2”

f) is this for dose 1 or dose 2? not specified in the figure or figure legend. What is boost 7? Elaborate in figure legend or make clearer in figure

g) hard to read, possible fig title “Plasma cells D1 vs D11 post dose 1”

16.

Lines 236-242: This data comes out of seemingly nowhere. It would be good to introduce more clearly why this particular gene and region was analyzed.

17.

Lines 238-240: Moreover, a particular CDRH3 amino acid (+1 hamming distance) 238 sequence was shared by several post-vaccination samples (13/79) that was not observed at baseline 239 (0/40) (Figure 4a-d). Is this actually describing Fig 4e?

18.

Figure 4: This figure is blurry and hard to see

a) Axes mixed up. Is the x axis actually the length, and the y axis the density, and the figure legend title “vaccination status”? Is this supposed to be before dose 1, 11 days after dose 1, and the combination of days post dose 2 (aka D7dose2 –it make more sense to say it like that instead)?

19.

Figure 5: Label top as dose 1 and bottom as dose 2 in figure

20.

Figure 5d-f not mentioned in text. Put in supplementary if it's not going to be mentioned.

21.

Line 276: “from 13 samples from 5 individuals” should this be “from 13 samples from 5 individuals collected on days 1, 11, and 64/92”

22.

Figure 6:

a) Please elaborate on the additional labels, arrows, squared data points, what APS 1 means (automatic population separator?). Is this PCA of FACS Ab prevalence or does this take into account the gene expression as well? Also state that these are day 256 cells

b) Which cells were used (1, 11, and 64/92? Or just day 1?).

c) Which cells are you using from which day?

d and e) What are the colors? What do they mean? Are they the 5 individuals?

h) The hamming distance of CDRH3 as an ebola binder post vaccination is further on average? If there is a subfraction that is closer, this needs to be boxed/highlighted/marked in some way and written about clearly in the legend and the main text

23.

Line 311: Pseudobulk of which samples? How long after first and second doses? (Referring to figure 7a, could also state in

figure legend).

24.

Line: 534 – What was cohort/group 3? Still unclear.

25.

Supplementary tables: Group 2 is repeated twice in all the tables instead of group 3 being written (under “Study day”)

Reviewer #3

(Remarks to the Author)

The manuscript by O'Connor et al. evaluates the B cell responses of human subjects vaccinated with a two-dose vaccine regimen against Ebola virus (Ad26.ZEBOV, MVA-BN-Filo). The authors perform a deep characterization of the B cell responses in vaccinees utilizing systems immunology approaches and demonstrate that prime and boost vaccination induce strong humoral responses including B cell memory. The authors also propose early biomarkers with predictive value to assess subsequent B cell immunity. Overall the study is well conducted. Some conclusions may not be fully supported by the data presented. Specific concerns are as follows

- 1- Please revise the text, some paragraphs have different fonts combined and there are quite a lot of misspelling errors. It also would have been helpful to add page and line numbering for reviewing purposes.
- 2- The methods section needs to add information on the tools utilized for machine learning and it reads like a lab protocol in some paragraphs, supplier informations are missing as well. Not all scripts for the different sequencing analyses are provided.
- 3- It is quite complicated to understand the tests performed for each cohort and group (despite the schematic shown in Figure 1) perhaps a more direct comparison between the UK data and African data indicating the time points compared would be more clear?
- 4- The fact that there are no B cell memory responses observed in the African cohort is quite puzzling. The authors propose one reason could be the presence of anti Ad26 antibodies in the cohort participants, which may be the case. Still I think it would be relevant to show a positive control. Can other B cell memory cells be identified in these samples? Do they actually have anti Ad26 Abs to a greater extent than the UK samples?
- 5- Regarding the DEG data, can the authors propose a set of genes related with the robustness of the humoral immune response to EBOV vaccination? Are there perhaps other data repositories that could help to evaluate whether the signatures observed are EBOV GP specific? For example, p21 and p27 are cell cycle regulators that are likely to be upregulated in response to any immune stimulus. In general, the authors tend to overinterpret the RNA sequencing data and scSeq differentially expressed genes (low Log2FC) and suggested implications (in the last section of the results and throughout the discussion).

Minor:

- Across the ms the authors should probably use the revised filovirus nomenclature using Orthoebolavirus zairensis to refer to the virus species and Ebola virus (EBOV) to describe the virus (Biedenkopf et al. 2023 10.1007/s00705-023-05834-2). Sometimes Ebola or Zaire alone are used which is not correct.
- Abstract “These results demonstrate that early immune responses —captured in blood using systems immunology approaches — are delineative and predictive of B cell responses to vaccination.” In what sense predictive?
- Ensure that text references match the figures they are referring to.
- Fig. 2: Change fold rise to fold change; subfigures are too small and not readable (b-d). Fig2a: explain the arrows (time of boost?) in the legend.
- “This motif is like a previously described EBOV GP binding monoclonal antibody — differing by two amino acids at position 5 and 6 (Rijal et al., 2019).” move to discussion
- Fig. 4: a-d) CDR3 labeled (not CDRH3 as in the legend?); e) legend too small, f) legend too small#
- Fig. 6: 2x d); h) change Ebolavirus to EBOV
- Supplementary figures too small: 1, 2, 7, 16, 19a-d, 14g (2nd 14), 16 (2nd 16)
- Discussion: clarify when talking about MVA-BN-Filo, MVA-vectored vaccines in general, and the empty MVA vector.
- “The minimum to maximum ranges of BMEM/million cultured PBMCs in EBL2001 vs EBL2002 were (Group 1: 0.10-42.50 vs 0.10-21.25) and (Group 2: 6.75-153.8 vs 0.10-10.10).” should go to the results section.

Version 1:

Reviewer comments:

Reviewer #1

(Remarks to the Author)

The authors have addressed all the review points that were brought up in the initial review process and it's the opinion of this reviewer that the manuscript should be accepted for publication.

Reviewer #2

(Remarks to the Author)

The authors have made a serious effort to improve the manuscript and have adequately addressed this reviewer's comments.

Reviewer #3

(Remarks to the Author)

My concerns have been addressed. I have no further comments on this manuscript

Reviewer #4

(Remarks to the Author)

This study investigates the human B-cell immune response following vaccination against Ebola virus disease using the Ad26.ZEBOV and MVA-BN-Filo vaccines, using data from two clinical trials—EBL2001 (UK) and EBL2002 (Africa). Employing bulk and single cell transcriptomic methods, the researchers analyzed plasma cell and memory B-cell responses post-vaccination. The authors employed a linear support vector regression (SVR) model to predict antibody responses based on differentially expressed genes (DEGs) measured 10 days after vaccination. Importantly, the study identified a unique B-cell receptor sequence (CDRH3) that emerged after vaccination, closely resembling known antibodies that bind specifically to the Ebola virus glycoprotein. Overall, the processing of both bulk RNA-seq and single-cell RNA-seq data appears standard and appropriate. However, we have concerns regarding specific data/methodological details and the potential for overinterpretation of some results. Our comments are below:

MAJOR COMMENTS:

- Study design - The study design is complex, involving multiple cohorts and randomized vaccine groups across two separate trials—EBL2001 (UK) and EBL2002 (Africa). Trial participants were recruited in cohorts. Participants in each cohort were randomized into three groups based on the timing of dose 2 (MVA-BN-Filo), to study the effect of dose interval on the B cell memory responses. The study design is very complex and could benefit from a simpler diagram, perhaps with each trial shown separately, and only including the parts relevant to the manuscript. Also, it appears that assay data from multiple vaccine groups were combined in the analyses, which introduces potential for confounding.
- Differential gene regulation coinciding with the peak of plasma cell responses following vaccines. The authors analyzed differentially expressed genes through bulk RNA sequencing of whole blood samples at the peak of plasma cell responses to the first dose of vaccine (Ad26.ZEBOV) and second dose of vaccine (MVA-BN-Filo). It is unclear which cohorts and groups were used for this particular analysis. If multiple groups were used, they may have different peak times. Additionally, figures 3a and 3b should be labeled as 'D11 vs D1 post dose 1' and 'D7 vs D1 post dose 2', since D1 is the baseline. Figure 3f – The figure legend should include the number of samples in each group.
- Machine learning model to predict antibody responses. The authors used the top 200 DEGs as input features for a linear SVR model trained on a small dataset (N = 28). First, it is unclear what specific feature values were used as model input (for example, normalized expression of the top 200 DEGs, or some transformed values). This should be clearly stated. Second, the model appears to perform perfectly on the training set (Figure 5a, 5d), but much worse on the validation set (Figure 5b, 5e), raising serious concerns about overfitting, especially given the high dimensionality relative to the small sample size. Third, the rationale for selecting a linear SVR over non-linear models is not discussed. Clarifying this choice would strengthen the analysis. Furthermore, have the authors checked correlation between gene expression with antibody levels? If the top predictive genes identified by the linear model are strongly correlated with antibody levels, it is unclear why a machine learning model is necessary at all, as a simpler correlation analysis or regression approach might be more interpretable and equally effective.
- Single cell analysis of B cells: In lines 315 - 317: authors state “relative frequency of plasma cells was observed for the majority of vaccinees at 10 days after dose 1 and 7 days after dose 2”, but figure only shows dose 1 data only.

In Figures 6d and 6e, the number of plasma and proliferating cells per donor and time point should be reported. Given the low frequency of plasma cells (~0.1% in some samples), providing cell number is important for assessing the reliability of relative frequency estimates.

Line 318 – 319 and Figure 6e: the authors claim that “a nominal statistical significance of decrease in proliferating B cells”, but Figure 6e has 4 groups, it's not clear which specific groups are being compared.

Figure 6f, 6g – The figure legend should include the number of samples in each group, statistical comparison method used and indicate whether the samples are paired or not. Also, have the p values been adjusted for multiple testing?

• DEG analysis based on pseudo bulk gene expression of plasma cells: As mentioned earlier, plasma cell cluster is rather small. Without cell numbers per donor and time point, it's hard to estimate reliability of DEG results (with too few cells, the result may be highly sensitive to noisy signals).

The figure legend of figure 7 should be clearer. For figure 7b and 7c, what are log2FC cutoffs the authors used? For figure 7d and 7e, how 'Down', 'NS', and 'UP' are defined? If NS means non-significant, why they are shown here? Can authors also show leading edge genes in each pathway (or include in supplementary tables)?

The authors may want to remove this section from the manuscript or discuss the limitations.

MINOR COMMENTS:

- Figure 2g is cited before Figure 2f.
- In Fig6c, the gene names are not visible.
- Lines 311–313 refer to Supplementary Figures 7a and 7b, but only a single Supplementary Figure 7 is provided
- Acronyms need to be defined in abstract “EBOV GP-binding antibodies”
- Typo in gene name: “but their impact on CHR3 structure and epitope interactions are not known”.
- Missing word: “Moreover, observed a cluster of highly similar CDRH3 sequences (within one hamming distance)”
- Page 9 Line 15: “used by the an EBOV GP binding monoclonal that is an edit distance of 2”

Reviewer #5

(Remarks to the Author)

Version 2:

Reviewer comments:

Reviewer #4

(Remarks to the Author)

The authors have addressed all comments, and the manuscript is now much clearer and easier to understand.

A few minor typos still exist:

- Line 328 "vaccinees".
- Fig 6 d and e: It would be helpful to label the groups the individuals are from.

Reviewer #5

(Remarks to the Author)

Point-by-point response

Reviewer #1 (Remarks to the Author):

Summary: The manuscript Phenotypic and transcriptomic characterization of the human B cell response to a novel Ebola vaccine regimen (Ad26.ZEBOV, MVA-BN-Filo) by O'Connor et al. utilized cellular, transcriptomic, and machine learning to fully characterize the B cell phenotypes stimulated and associated follicular T cell populations that contribute to the B cell population formation, following heterologous Ad26.ZEBOV and MVA-BN-Filo vaccination in two clinical cohorts. The authors were able to identify a transcriptomic signature that correlated with magnitude of antibody responses. Deeper analysis into the BCR repertoire revealed a unique CDRH3 sequence that is similar to previously described antibody sequences. The authors utilized this unique sequence to further characterize the B cell cellular populations associated with this unique CDRH3 sequence post vaccination. This study not only describes the unique B cell repertoire, but it also compares the similarities of the stimulated immune response with clinical cohorts in Europe and Africa to ensure the validity of the described results in the effected country. Overall, a very through transcriptomic approach to B cell characterization in multiple human cohorts demonstrating strong associations across cohorts to ensure predictive power of their model system moving forward. My suggestion is for minor revisions before being accepted for publication.

Major comments:

1. Is there any way for the authors to include archived data from a homologous vaccination clinical cohort to delineate differences in the heterologous vaccination strategy?

While there are no studies explicitly addressed B cell subset-specific responses (e.g., plasma cells, memory B cells) or utilised RNA sequencing to analyse transcriptional changes in the context of homologous Ad26.ZEBOV regimens, humoral responses to homologous regimens have been evaluated. We have included reference to this in the limitations section of discussion as below:

Page 15, lines 40–44— *Homologous 2-dose regimens (e.g., Ad26, Ad26 or MVA, MVA) are less immunogenic than heterologous strategies but induce humoral immune memory (Goldstein 2022). Since homologous 2-dose regimens have not been examined at the resolution of B cell subset-specific responses (e.g., plasma cells, memory B cells) or through RNA-seq analysis, direct comparison with the data presented here is not possible.*

2. In the unique CDHR3 sequencing key amino acids 5 and 6 have multiple sub variations possible can the authors comment on the impact of the less common amino acid sequences and how they could potential contribute.

The impact of these substitutions on epitope interactions is shaped by both the nature of the substitution and the structural context. Determining the precise effects of these amino acid differences would require experimental validation, such as binding assays and structural analyses, which are beyond the scope of this study.

However, in functional antibodies, amino acids (a.a.) at the terminal positions of the CDRH3 are typically more conserved, whereas those in the central positions tend to exhibit greater diversity (Mejias-Gomez, 2023). Of note, a previously described Ebola virus glycoprotein binding monoclonal antibody differs from the motif by two amino acids at position 5 and 6. This has been mentioned in the discussion section:

Page 15, lines 31-38 — *Moreover, observed a cluster of highly similar CDRH3 sequences (within one hamming distance) that were present exclusively in post-vaccination samples. This CDRH3 sequence cluster contained the IGHV3-15 variable gene and its motif differed by only two amino acids, at positions 5 and 6, from a known Ebola virus glycoprotein binding antibody (Rijal et al., 2019). Central CDRH3 amino acid positions, such as position 5 and 6, a tend to exhibit greater diversity but their impact on CHRH3 structure and epitope interactions are not known (Mejias-Gomez et al., 2023). However, CDRH3 sequences within this motif have been observed following Ebola virus disease (Davis et al., 2019).*

3. Can the authors use the BCR sequencing as an additional filter to determine antigen specificity during the analysis?

Unambiguously determining antigen-specificity would require pair heavy and light chain data and then experimental validation, such as binding assays and structural analyses, which are beyond the scope of this study. However, we have provided some circumstantial evidence of specificity.

Page 15, lines 31-38 — *Moreover, observed a cluster of highly similar CDRH3 sequences (within one hamming distance) that were present exclusively in post-vaccination samples. This CDRH3 sequence cluster contained the IGHV3-15 variable gene and its motif differed by only two amino acids, at positions 5 and 6, from a known Ebola virus glycoprotein binding antibody (Rijal et al., 2019). Central CDRH3 amino acid positions, such as position 5 and 6, a tend to exhibit greater diversity but their impact on CHRH3 structure and epitope interactions are not known (Mejias-Gomez et al., 2023). However, CDRH3 sequences within this motif have been observed following Ebola virus disease (Davis et al., 2019).*

Minor Comments:

1.) Zaire ebolavirus is old nomenclature adjust to current nomenclature.

Updated to new nomenclature: *Orthoebolavirus zairense*, *Orthoebolavirus sudanense*, *Orthomarburgvirus marburgense*, and the *Orthoebolavirus taiense*

2.) At the end of the introduction emphasize the samples are from a human clinical cohort don't downplay the importance of your samples.

Thank you for this comment, we have edited the end of the introduction as follows. Page 2, line 38-40— *This clinical study provides an unprecedented insight into the generation and maintenance of the long-lasting B cell immunity following Ebola vaccination of humans.*

3.) Result section one: change follow-on study to follow-up study.

Corrected

4.) Figure 2 and Supplemental font different.

Figure 2 and supplemental figure 15 font differences have been addressed and the text has been edited to be more legible.

5.) Remove the syringe art in figure 1.

This figure has been updated, clip art removed and more clarity given to Studies, Cohorts and Groups.

6.) Supplemental 16-18 reference font different.

Figures have been edited to improve legibility and edit font.

7.) Keep day abbreviation vs writing it out consistent throughout the manuscript to goes back and forth.

In the text day is written out

8.) Discussion correct centres to centers and titres to titers.

Edited these to American English.

Reviewer #2 (Remarks to the Author):

BRIEF SUMMARY:

O'Connor / Clutterbuck et al. have investigated the characteristics of Ad26.ZEBOV, MVA-BN-Filo vaccine immune responsiveness from UK/Africa clinical trial samples in a way that attempts to go into more depth than other previous publications using these same clinical trial samples. They assess gene expression and abundance changes of B cells, T follicular helper cells and plasma cells over time with FACS and EliSpot after the first and second vaccine dose. They determine differentially expressed genes of participants after the first and second dose with bulk RNA-seq. They also use bulk RNA-seq alignments to characterize changes, after the first and second dose, in B cell receptor CDRH3 length and sequence, finding a motif after vaccination that is similar one that has been implicated in Ebola virus glycoprotein binding. Furthermore, they developed an algorithm to predict vaccine immune responsiveness, using machine learning that is trained on the RNA-sequencing data to find the top genes that best predict this responsiveness. Finally, they used single-cell analyses with FACS cell-type enrichment and PCA/UMAP with single cell RNA-sequencing, assessing gene expression of different cell types and changes in cell-type/subtype abundance and gene expression after vaccination.

Understanding an immune system's response to vaccination is incredibly important, and particularly for immunization against the Ebola virus, which is unfortunately too common in some African countries, highly contagious, and deadly.

OVERALL ASSESSMENT:

While the topic of this manuscript is highly important, the content is not distinctively novel or cohesive, and it lacks comprehensiveness. The manuscript may not yet meet the standard expected for a publication in Nature Communications. Significant revision is needed, including more detailed descriptions, enhanced detail, and improved organization. I hope the following comments will be of help to improve the manuscript.

MAJOR COMMENTS:

1.

Lines 79-90: Should mention the staggered timing of the 2nd dose in the manuscript text and a brief rationale for testing this variable.

We have added a sentence here to clarify the inclusion of different dosing regimens into the trial.

Page 2: Lines 13-15: *The spacing of the doses was compared to determine the optimal spacing for enhancing the immunogenicity outcomes and to have some flexibility of dosing in the field.*

2.

Lines 82-88. You describe Cohorts 1 and 2 but not Cohort 3. Are “cohorts” the same thing as “groups”?

We have added explanation to the Figure 1 legend and added to the main body of text to clarify the difference between Cohort and Group.

Page 3, Lines 4-9: *Within the EBL2001 and EBL2002 trials, participants were recruited in Cohorts. For EBL2001 in the UK a sentinel Cohort 1 (n=30) was recruited specifically to assess peak plasma cell time points for sample collection in the main trial Cohorts 2+3. Cohort 2 was the exploratory cohort for collection of PBMCs and DNA samples, while Cohort 3 was for main clinical trial objectives only. Each cohort was randomised into Group 1, Group 2 or Group 3 depending on when they would receive the MVA-BN-Filo (dose 2). For EBL2002, samples were only available from Cohort 2 (vaccine Groups 1 and 2).*

3.

Lines 236-242: Observations/conclusions for Figure 4a, 4b, 4c, 4d should be stated in the manuscript.

We have described expanded the description of these figures in the results section as well as the interpretation in the discussion section.

Page 8, line 7–12 — *We extracted the B cell receptor sequences from the whole blood RNA-seq data and described study time point dependent differences in B cell repertoire properties: CDR3 length, insert size, VJ junction size and convergence (Figure 4a-d). The CDRH3 length distribution shifted, and the VJ junction size increased 10 days post-vaccination compared with baseline (Figure 4a, c). Additionally, both vaccine doses led to an increase in the mean CDRH3 insert size and convergence relative to baseline (Figure 4a, c).*

Page 14, line 27– 31— *We extracted the B cell receptor sequences from the whole blood RNA-seq data. Post-vaccination, we observed a shift in the CDRH3 length distribution and an increase in both the mean CDRH3 insert size and VJ junction size, reflecting an adaptive response aimed at generating a broader repertoire of antigen-specific BCRs (Galson et al., 2015). Additionally, the convergence of CDRH3 sequences post-vaccination suggests the selective expansion of vaccine-specific B cell clones.*

4.

Missing from methods:

Differential expression analyses – how? what script/package/parameters?

Description of the differentially expression analysis is now included as below:

Page 22 line 10–17 —

Gene expression analysis

Differential gene expression analysis was performed using the R Bioconductor packages edgeR and limma (Robinson et al., 2010, R core team, 2013, McCarthy et al., 2012) RNA-sequencing data were normalized for RNA composition using the trimmed mean of M-value (TMM) method (Robinson and Oshlack, 2010) (Robinson & Oshlack, 2010). The data were then transformed using the voom function in limma. A linear model was fitted to the data with the lmFit function, employing the empirical Bayes method to share information across genes (Ritchie et al., 2015). Paired analyses were conducted to compare pre- and post-vaccination samples at each study time point, with a statistical significance threshold set at a false discovery rate (FDR) < 0.05.

blood transcriptional module (BTM) analysis – how? what script/package/parameters?

Description of the BTM analysis is now included as below:

Page 22 line 18–22 — *Blood transcriptional module analysis was performed using the "tmod" R package, with genes ranked based on their log-ratio (LR) values.*

Module expression was assessed using the "tmodCERNOtest" function, a non-parametric statistical test that operates on gene ranks (Weiner, 2016).

CIBERSORTx – how was it use? parameters?

Description of the CIBERSORTx analysis is now included as below:

Page 22 line 32–37 —

Cell abundance deconvolution from whole blood transcriptomic data

The cell composition of whole blood samples was assessed using the CIBERSORTx method (Newman et al., 2019). A filtered, non-log-transformed reads per kilobase million (RPKM) sample gene matrix, along with a "signature" gene file (LM22; representing 22 immune cell types), was used to deconvolute cell abundances.

CIBERSORTx was executed in relative mode with B-mode batch correction and 1,000 permutations.

CDR3 length, insert size, ndn size, convergence, motif analyses – how were these analyses done after the Mixcr output of the receptor sequence?

Summary of the CDR3 analysis is included as below:

Page 22 line 38–42—

B cell receptor profiling from bulk RNA-sequencing data

B cell receptor sequence information was extracted from bulk RNA-sequencing using the MiXCR (MIXCR (version 2.1.12) functions "Align," "Assemble partial,"

"Assemble," and "Export clones" using default settings (Bolotin et al., 2017). Basic statistics were calculated on MiXCR output using VDJtools (version 1.1.10) (Shugay et al., 2015).

The detailed analysis code for all the above methods (comment 4) is now deposited on GitHub (https://github.com/dan-scholar/Ebola_vaccine_B_cell.git). This is now mentioned as below:

Page 22, line 6-7— *The analysis script is deposited on GitHub (https://github.com/dan-scholar/Ebola_vaccine_B_cell.git).*

5.

Line 666: Nothing is written under “Machine learning derived predictors of immune responses”. How was this done? Did you build it off of a previous study? Explain your methodology in detail.

Apologies for this omission, details now include as below:

Page 22, line 24–30 —

Machine learning derived predictors of immune responses

We developed a supervised machine learning algorithm to predict GP-specific antibody levels based on changes in differentially expressed genes post-vaccination. A linear kernel support vector regression (SVR) model was trained on 75% of the data, using 5-fold cross-validation repeated 10 times to optimize performance. The trained model's performance was then evaluated on the remaining 25% of the dataset. The full analysis script is deposited on GitHub (https://github.com/dan-scholar/Ebola_vaccine_B_cell.git).

MINOR COMMENTS (AND SUGGESTIONS):

6.

Introduction - lack of citation in the introduction for clinical trials and other publications, eg.

[https://doi.org/10.1016/S1473-3099\(24\)00058-6](https://doi.org/10.1016/S1473-3099(24)00058-6) — This citation is now included on page 2, line 33–35 — *Rapid recall responses have been described after a Ad26.ZEBOV booster vaccination of individuals who received a primary Ad26.ZEBOV, MVA-BN-Filo Ebola vaccine regimen (Larivière et al., 2024).*

<https://doi.org/10.1371/journal.pmed.1003865> — this citation is now included on page 1, line 38 – page 2, line 3 — *A second vaccine Ad26.ZEBOV (Zabdeno, Janssen) used with MVA-BN®-Filo (Mvabea, Janssen) has demonstrated safety and immunogenicity in healthy volunteers, including down to 1 years of age, with evidence for efficacy being inferred from immunobridging studies (Afolabi et al., 2022, Bockstal et al., 2022a, Anywaine et al., 2022).*

<https://doi.org/10.3389/fimmu.2023.1215302> — This citation is now included on page 2, line 31-33— *Vaccine-induced immune memory has been proposed as a correlate of protection, as the anamnestic response can potentially halt disease progression during the extended viral incubation period (McLean et al., 2023).*

<https://doi.org/10.1080/21645515.2017.1264755> — This citation is now included on page 2, line 4-10— *A phase 1 clinical trial (EBL1001) in the UK compared dosing order and spacing of two vaccines: Ad26.ZEBOV (adenovirus serogroup 26 vaccine encoding Ebola Mayinga GP, dose 1) and MVA-BN-Filo (modified vaccinia Ankara encoding GPs from EBOV, Orthoebolavirus sudanense (SUDV), Orthomarburgvirus marburgense (MARV), and the Orthoebolavirus taiense (TAFV) nucleoprotein, dose 2). In this trial, 97% of participants receiving Ad26.ZEBOV as dose 1 had detectable*

Ebola GP IgG responses by day 28, and all participants had detectable IgG responses 21 days post-boost, with responses still detectable after 8 months (Shukarev et al., 2017).

7.

Figure 1:

Can get quite messy when the staggered timings and the “common” timings are all put together. Could be helpful to visually distinguish them (maybe a box around the “common” timings).

We have updated figure 1 to clarify the Clinical Trials, study cohorts, vaccine groups and assays. Boxes with hatched outlines have been added to the table to indicate where time points are shared between studies, for each assay.

8.

Figure 2:

For easier read, could be helpful to write dose 1 and dose 2 under the figure panels e-g to distinguish, or specify dose 1 (left), dose 2 (right) in legend

f) Helpful to write more precisely in legend what surface markers mean, e.g., subtype CD86 (activated), subtype CD62L (ag specific migratory), and $\alpha 4\beta 1$ pos (enhanced survivability)

We have updated figure 2 to clarify these points and the legend has been updated.

9.

Line 134: Mistake in Figure 2 legend

e) Ex vivo ELISpot Frequency of Total IgG-ASC and g) EBOV-GP specific IgG-ASC should be

e) Ex vivo ELISpot Frequency of Total IgG-ASC (top) and EBOV-GP specific IgG-ASC (bottom)

Figure 2e Legend: This correction has been made to the text and figure reference updated.

10.

Line 158: 2f cited before 2e in the paper – “Figures and tables in the main manuscript must be cited in the order they appear in the text, figure legends, table legends and boxes”

This sentence has been corrected to refer to information in the correct order.

11.

Line 176-180: Transient increase in migratory, vaccine specific plasma cells can be used to detect the presence of ongoing vaccine responses with surface markers such as CD62L, which is indicative of antigen specific migratory plasma cells (Mei et al., 2009), CD86, which is highly expressed on recently activated plasma cells (Rudolf-Oliveira et al., 2018) and $\alpha 4\beta 1$ (VLA-4) which enhances bone marrow survival (Khodadadi et al., 2019).

This paragraph has been moved from the results section to the discussion to introduce the plasma cell characterisation [page 17, lines 30-34]

12.

Line 189: Add: While these findings are not statistically significant, they do point to a

trend of plasma cell responses to both dose 1 and dose 2 administration.

This sentence has been added to the results section.

Page 6, line 39-41— *While these findings are not statistically significant, they do point to a trend of plasma cell responses to both dose 1 and dose 2 administration.*

13.

Line 193: Change “We next explored gene regulation in whole blood samples” to “We next explored gene regulation with bulk RNA sequencing of whole blood samples”

Edited as suggested page 6, line 45

14.

Line 197: sixteen genes not “sixteen gene”

Edited as suggested page 6, line 39

15.

Figure 3:

a) DEG dose 1 –change title to “D1 vs D11 post dose 1”

Edited as suggested

b) DEG dose 2 – change title to “D1 vs D7 post dose 2”

Edited as suggested

c) Colors for points feel arbitrary– could have (a) title be one color, (b) title be another color, then these refer to the points in c

Thank you we have considered this suggestion. However, the colours of the points in a) and b) reflect up (red) and down (blue), which is consistent across plots so changing this would lose this consistent thread.

d) Organization of modular signatures – is it from most to least enriched?

Alphabetical? Largest to smallest group of genes?

In this plot we have used the default organisation of modular signatures – which is ordered by qval. I have include this in the figure legend for clarity

Figure 3 — d) *Modular signatures induced during different study time points, enriched modules ($FDR < 1 \times 10^{-3}$) are displayed order by q-value.*

e) Title “D11 post dose 1 vs D7 post dose 2”

Edited as suggested

f) is this for dose 1 or dose 2? not specified in the figure or figure legend. What is boost 7? Elaborate in figure legend or make clearer in figure

Edited to make clear which timepoint/dose i.e., D1, D11 (dose 1) and D7 (dose 2)

g) hard to read, possible fig title “Plasma cells D1 vs D11 post dose 1”

Edited to increase font size to make easier to read

16.

Lines 236-242: This data comes out of seemingly nowhere. It would be good to introduce more clearly why this particular gene and region was analyzed.

We have introduced the rationale for examining the CDR3 as below:

Page 9, line 3-7 — *The upregulation of a shared set of genes encoding antibody segments following both vaccine doses prompted an investigation into whether this phenomenon reflected convergent, antigen-specific B cell receptor sequences. Given that immunoglobulin specificity is primarily determined by the hypervariable*

region (CDR3), we analysed CDR3 repertoire information extracted from the bulk RNA-sequencing data using the MiXCR software platform (Bolotin et al., 2017).

17.

Lines 238-240: Moreover, a particular CDRH3 amino acid (+1 hamming distance) 238 sequence was shared by several post-vaccination samples (13/79) that was not observed at baseline 239 (0/40) (Figure 4a-d). Is this actually describing Fig 4e?

Yes – this is what is shown in Figure 4e. For the datapoint to be visible, it was necessary to make them non-overlapping. We have inserted a horizontal line to make it clear that data points below this are zero. This is not described in the figure legend as below:

Figure 4 legend— e) Identification of a CDRH3 sequence (and sequences within 1 hamming distance of this sequence) exclusively seen post-vaccination. Data points below the horizontal dashed line are 0.

18.

Figure 4: This figure is blurry and hard to see

a) Axes mixed up. Is the x axis actually the length, and the y axis the density, and the figure legend title “vaccination status”? Is this supposed to be before dose 1, 11 days after dose 1, and the combination of days post dose 2 (aka D7dose2 –it make more sense to say it like that instead)?

Corrected figure 4a axis labels. I have updated the legend for 4a as suggested to D1, D11 (dose 1) and D7 (dose 2). Higher definition figures have also been included.

19.

Figure 5: Label top as dose 1 and bottom as dose 2 in figure

Edited as suggested

20.

Figure 5d-f not mentioned in text. Put in supplementary if it's not going to be mentioned.

These figures are now mentioned as below:

Page 10, line 6–9— A machine learning model based on the most differentially expressed genes at 7 days post-second vaccine dose successfully predicted specific antibody responses to this dose (Figure 5d & 5e). The genes with the highest importance in this predictive model are highlighted in Figure 5f.

21.

Line 276: “from 13 samples from 5 individuals” should this be “from 13 samples from 5 individuals collected on days 1, 11, and 64/92”

Edited as suggested

22.

Figure 6:

a) Please elaborate on the additional labels, arrows, squared data points, what APS 1 means (automatic population separator?). Is this PCA of FACS Ab prevalence or does this take into account the gene expression as well? Also state that these are day 256 cells

We have added text to the legend and reference to Supplementary Figure 19 which describes the gating strategy used to inform the APS plot generation.

Figure 6a legend — *APS-1 (automatic population separator) of the flow cytometric data based on the expression of surface markers listed in the methods, Table ii, Panel A and Supplementary figure 19). The square symbols indicate each individual, coloured by population, and the contour lines indicate 1xStandard Deviation for each population. The dotted arrow indicates the direction of differentiation and the solid arrow the direction of plasma cell differentiation.*

b) Which cells were used (1, 11, and 64/92? Or just day 1?).

These are cells from all timepoints — now described in legend.

c) Which cells are you using from which day?

These are cells from all timepoints — now described in legend.

d and e) What are the colors? What do they mean? Are they the 5 individuals?

Each coloured line is an individual and this is now described in the legend.

h) The hamming distance of CDRH3 as an ebola binder post vaccination is further on average? If there is a subfraction that is closer, this needs to be boxed/highlighted/marked in some way and written about clearly in the legend and the main text

Thank you for this comment we have highlighted this subfraction with a blue arrow and mentioned in text and legend.

Page 11, line 9-11, — *The minimal number of amino acid changes needed to match a known GP binding mAb was three, this was the case for two plasma cell CDRH3 sequences both seen post-vaccination (Figure 6h – highlighted with blue arrow)*

Figure 6 legend — The two plasma cell CDRH3 sequences closest to known GP binding monoclonal antibodies (within 3 amino acid changes) are highlighted with a blue arrow.

23.

Line 311: Pseudobulk of which samples? How long after first and second doses? (Referring to figure 7a, could also state in figure legend).

This pseudobulk analysis on cells from all timepoints but each time point is represented by a different shape in the plot. This has been made clearer in the figure legend. Triangle for baseline, filled square for first dose +10 days and filled circle for second dose + 7 days.

24.

Line: 534 – What was cohort/group 3? Still unclear.

Thank you for this comment this has been be clarified as per comment 2.

25.

Supplementary tables: Group 2 is repeated twice in all the tables instead of group 3 being written (under “Study day”)

This has been corrected for all tables.

Reviewer #3 (Remarks to the Author):

The manuscript by O'Connor et al. evaluates the B cell responses of human subjects vaccinated with a two-dose vaccine regimen against Ebola virus (Ad26.ZEBOV, MVA-BN-Filo). The authors perform a deep characterization of the B cell responses in vaccinees utilizing systems immunology approaches and demonstrate that prime and boost vaccination induce strong humoral responses including B cell memory. The authors also propose early biomarkers with predictive value to assess subsequent B cell immunity. Overall the study is well conducted. Some conclusions may not be fully supported by the data presented. Specific concerns are as follows

1- Please revise the text, some paragraphs have different fonts combined and there are quite a lot of misspelling errors. It also would have been helpful to add page and line numbering for reviewing purposes.

Font inconsistencies have been addressed.

2- The methods section needs to add information on the tools utilized for machine learning and it reads like a lab protocol in some paragraphs, supplier informations are missing as well. Not all scripts for the different sequencing analyses are provided.

Greater detail of the machine learning analysis is now provided, and script is uploaded on GitHub

Page 22, line 24-30 —

Machine learning derived predictors of immune responses

We developed a supervised machine learning algorithm to predict GP-specific antibody levels based on changes in differentially expressed genes post-vaccination. A linear kernel support vector regression (SVR) model was trained on 75% of the data, using 5-fold cross-validation repeated 10 times to optimize performance. The trained model's performance was then evaluated on the remaining 25% of the dataset. The full analysis script is deposited on GitHub (https://github.com/dan-scholar/Ebola_vaccine_B_cell.git).

3- It is quite complicated to understand the tests performed for each cohort and group (despite the schematic shown in Figure 1) perhaps a more direct comparison between the UK data and African data indicating the time points compared would be more clear?

The figure 1 has been updated to clarify these differences. Boxes have been added to highlight the time points at which BMEM can be compared between the Clinical Trials.

4- The fact that there are no B cell memory responses observed in the African cohort is quite puzzling. The authors propose one reason could be the presence of anti Ad26 antibodies in the cohort participants, which may be the case. Still I think it would be relevant to show a positive control. Can other B cell memory cells be identified in these models? Do they actually have anti Ad26 Abs to a greater extent than the UK samples?

Page 15, Line 2-7: We have updated the discussion to say

... which showed equivalent IgG GMCs in comparison to those seen in the UK trial (Pollard et al., 2020). Despite the difference in Ebola specific memory B cell

responses, we showed that total IgG-memory B cell responses were similar between all populations at all time points (Supplementary Figure 15a). A direct comparison between the study sites in the UK compared to the African sites was not made as the studies were not designed for this purpose.

We have also updated supplementary figure 15 to show the total IgG-memory B cell frequency between study populations (Fig 15a).

5- Regarding the DEG data, can the authors propose a set of genes related with the robustness of the humoral immune response to EBOV vaccination? Are there perhaps other data repositories that could help to evaluate whether the signatures observed are EBOV GP specific? For example, p21 and p27 are cell cycle regulators that are likely to be upregulated in response to any immune stimulus. In general, the authors tend to overinterpret the RNA sequencing data and scSeq differentially expressed genes (low Log2FC) and suggested implications (in the last section of the results and throughout the discussion).

In this work, we have identified specific genes associated with the robustness of the humoral immune response, supported by orthogonal data as detailed in the examples below:

Page 10, lines 1-3— *A machine learning model using the 200 genes most differentially expression 10 days after the first dose of vaccine was able to predict specific antibody responses (Figure 5a &b). Analysis of the genes important in building this model included IGHV3-15 (ranked first in importance) and IGLV1-40 (ranked 11th in importance), these are the variable heavy and light chain genes used by the an EBOV GP binding monoclonal that is an edit distance of 2 from the CDRH3 uniquely seen post-vaccination in this data set (Figure 4f & Figure 5c).*

&

Page 15, lines 23-26 — *Interestingly, IGHV3-15 and IGLV1-40 were ranked amongst the most important genes in this predictive model, and this variable heavy and light chain pairing is the most common combination in the publicly-available EBOV specific antibody sequences (Rijal et al., 2019).*

Evaluating the specificity of other genes, such as p21 and p27, is more challenging, as you pointed out, due to the lack of additional repositories that could further validate these relationships. However, we addressed this by splitting our machine-learning model into training and test sets to enhance the rigor of our analysis.

Minor:

- Across the ms the authors should probably use the revised filovirus nomenclature using Orthoebolavirus zairensis to refer to the virus species and Ebola virus (EBOV) to describe the virus (Biedenkopf et al. 2023 10.1007/s00705-023-05834-2). Sometimes Ebola or Zaire alone are used which is not correct.

Corrected

- Abstract “These results demonstrate that early immune responses —captured in

blood using systems immunology approaches — are delineative and predictive of B cell responses to vaccination.” In what sense predictive?

The abstract has been edited and this sentence highlighted has been removed and replaced with the following:

Page 1 line 21-22 – *Machine-learning models trained on blood gene expression predicted antibody response magnitude.*

- Ensure that text references match the figures they are referring to.

Linking of text references to the relevant figures has been checked.

- Fig. 2: Change fold rise to fold change; subfigures are too small and not readable (b-d). Fig2a: explain the arrows (time of boost?) in the legend.

Fold rise has been edited to fold change in text and figures.

Reference to the meaning of the arrows has been added to the legend for this figure.

- “This motif is like a previously described EBOV GP binding monoclonal antibody — differing by two amino acids at position 5 and 6 (Rijal et al., 2019).“ ♦ move to discussion

Moved as suggested.

- Fig. 4: a-d) CDR3 labeled (not CDRH3 as in the legend?); e) legend too small, f) legend too small#

Corrected labels and increase legend font size as suggested.

- Fig. 6: 2x d); h) change Ebolavirus to EBOV

Corrected

- Supplementary figures too small: 1, 2, 7, 16, 19a-d, 14g (2nd 14), 16 (2nd 16)

We have increased the size/font these figures as requested: 1, 2, 7, 14, 16 and 19

- Discussion: clarify when talking about MVA-BN-Filo, MVA-vectored vaccines in general, and the empty MVA vector.

Clarified as below:

Page 14, line 35-44 — *This increase in activated cTfh cells post-first dose but not post-second dose is consistent with findings from other studies assessing these cells in an Ad followed by MVA (ChAd63-MVA) heterologous vaccination schedule (Nielsen et al., 2021). In mice, MVA (MVA-OVA) appears to be a weak inducer of GC B cells (Wang et al., 2016). Clinical studies have detected rises in cTfh cells when a priming MVA dose is given but not after a repeated dose (Anderson et al., 2020). This may suggest that when MVA is given as a second dose it promotes differentiation of existing memory B cells into plasma cells rather than GC re-entry (Akkaya et al., 2020). Nevertheless, we described boosting of BMEM after MVA-BN-Filo given as a second dose of vaccine, which is consistent with the literature of MVA-responsive IgG-secreting memory B cells following MVA (empty MVA) vaccine (Anderson et al., 2020).*

Page 16, line 20-21 — *IGHV3-53 has previously been shown to be upregulated following MVA (MVA.HIVconsv), suggesting this observation may be vector specific (Oriol-Tordera et al., 2022).*

- “The minimum to maximum ranges of BMEM/million cultured PBMCs in EBL2001 vs EBL2002 were (Group 1: 0.10-42.50 vs 0.10-21.25) and (Group 2: 6.75-153.8 vs 0.10-10.10).” should go to the results section.

We have moved this text to the results section

Page 3, lines 32-35 — *The minimum to maximum ranges of BMEM/million cultured PBMCs in EBL2001 vs EBL2002 were (Group 1: 0.10-42.50 vs 0.10-21.25) and (Group 2: 6.75-153.8 vs 0.10-10.10).*

REVIEWER COMMENTS

Reviewer #1 (Remarks to the Author):

The authors have addressed all the review points that were brought up in the initial review process and it's the opinion of this reviewer that the manuscript should be accepted for publication.

Thank you

Reviewer #2 (Remarks to the Author):

The authors have made a serious effort to improve the manuscript and have adequately addressed this reviewer's comments.

Thank you

Reviewer #3 (Remarks to the Author):

My concerns have been addressed. I have no further comments on this manuscript

Thank you

Reviewer #4 (Remarks to the Author):

This study investigates the human B-cell immune response following vaccination against Ebola virus disease using the Ad26.ZEBOV and MVA-BN-Filo vaccines, using data from two clinical trials—EBL2001 (UK) and EBL2002 (Africa). Employing bulk and single cell transcriptomic methods, the researchers analyzed plasma cell and memory B-cell responses post-vaccination. The authors employed a linear support vector regression (SVR) model to predict antibody responses based on differentially expressed genes (DEGs) measured 10 days after vaccination. Importantly, the study identified a unique B-cell receptor sequence (CDRH3) that emerged after vaccination, closely resembling known antibodies that bind specifically to the Ebola virus glycoprotein. Overall, the processing of both bulk RNA-seq and single-cell RNA-seq data appears standard and appropriate. However, we have concerns regarding specific data/methodological details and the potential for overinterpretation of some results. Our comments are below:

MAJOR COMMENTS:

- Study design - The study design is complex, involving multiple cohorts and randomized vaccine groups across two separate trials—EBL2001 (UK) and EBL2002 (Africa). Trial participants were recruited in cohorts. Participants in each cohort were randomized into three groups based on the timing of dose 2 (MVA-BN-Filo), to study the effect of dose interval on the B cell memory responses. The study design is very complex and could benefit from a simpler diagram, perhaps with each trial shown separately, and only including the parts relevant to the manuscript. Also, it appears that assay data from multiple vaccine groups were combined in the analyses, which introduces potential for confounding.

- We have redesigned figure 1 to show more clearly the study design for each cohort and which analyses were performed in each of the studies and on which timepoint samples the analyses were undertaken
- We have also updated the text in the results to incorporate these changes and provide more clarity on the study structure.
 - Results section, Page 3, lines 4-45— *Within the EBL2001 and EBL2002 trials, participants were recruited in Cohorts (Figure 1). For EBL2001, in the UK, a sentinel Exploratory Cohort 1 (n=30) was recruited specifically to assess peak plasma cell time points, following dose 1 and dose 2, to target sample collection in Cohort 2. Cohort 2 was the exploratory cohort for collection of PBMCs and DNA samples, while Cohort 3 (not discussed in this manuscript) was for main clinical trial objectives only (Pollard et al., 2020). Each study Cohort was randomised into Group 1, Group 2 or Group 3 depending on when they would receive the MVA-BN-Filo (dose 2). For EBL2002 (Kenya, Burkina Faso and Uganda), samples were only available from Cohort 2 (vaccine Groups 1 and 2). Sampling timepoints and vaccination regimens are shown in Figure 1, for each Study, Cohort and Group, along with timepoints tested for each analysis. In the UK trial, EBL2001 (EVOLVE), participants were recruited from the Oxford area. Cohort 1, consisted of (n=30) healthy adults aged 18-65 years old, and was recruited specifically, with a more intense sample schedule, for determination of the optimal plasma cell timing over 14 days post dose 1 and 9 days post dose 2. Cohort 2 (n=50 healthy adults aged 18-65 years) was recruited for immunogenicity, along with exploratory analysis of the B cell response via ELISpot, flow cytometry and transcriptomics, and included long term follow-up to day 265. A*

further study, PRISM, recalled participants from Cohort 2 at four years post MVA-BN-Filo (V1), to look for long-term maintenance of immune memory (manuscript in preparation). For memory B cell responses, six participants in each vaccine Group (G1, G2 and G3) had PBMCs available at V1 (4 years post dose 2) and V2 (4.5 year post dose 2). For analysis of BMEM responses in none-UK participants, PBMC samples were obtained from the EBL2002 trial sites in Kenya, Uganda and Burkina Faso. Only PBMCs from participants in vaccine Group 1 (n=8) and Group 2 (n=11) were available.

- Differential gene regulation coinciding with the peak of plasma cell responses following vaccines. The authors analyzed differentially expressed genes through bulk RNA sequencing of whole blood samples at the peak of plasma cell responses to the first dose of vaccine (Ad26.ZEBOV) and second dose of vaccine (MVA-BN-Filo). It is unclear which cohorts and groups were used for this particular analysis. If multiple groups were used, they may have different peak times. Additionally, figures 3a and 3b should be labeled as 'D11 vs D1 post dose 1' and 'D7 vs D1 post dose 2', since D1 is the baseline. Figure 3f – The figure legend should include the number of samples in each group.
- We have clarified in Figure 1 the cohort/groups that contributed to the bulk and single cell (10x) RNA sequencing.
- We have edited the labels of figure 3a and 3b as suggested and also include an arrow to clarify the directionality of change (i.e., up/down post-vaccination).
- We have included the number of participants in the Figure 3f. We also removed the individuals who received the placebo vaccine from the plot — this hasn't changed the interpretation but is more consistent with the rest of the results (e.g., Figure 3a and b etc).
- Machine learning model to predict antibody responses. The authors used the top 200 DEGs as input features for a linear SVR model trained on a small dataset (N = 28). First, it is unclear what specific feature values were used as model input (for example, normalized expression of the top 200 DEGs, or some transformed values). This should be clearly stated. Second, the model appears to perform perfectly on the training set (Figure 5a, 5d), but much worse on the validation set (Figure 5b, 5e), raising serious concerns about overfitting, especially given the high dimensionality relative to the small sample size. Third, the rationale for selecting a linear SVR over non-linear models is not discussed. Clarifying this choice would strengthen the analysis. Furthermore, have the authors checked correlation between gene expression with antibody levels? If the top predictive genes identified by the linear model are strongly correlated with antibody levels, it is unclear why a machine learning model is necessary at all, as a simpler correlation analysis or regression approach might be more interpretable and equally effective.
- We have clarified the specific feature values in the model in the methods section.
 - Page 23; lines 1-11— “A support vector regression (SVR) model was developed to predict antibody levels following the first vaccine dose. The dependent variable comprised the residuals of log₁₀-transformed ELISA measurements, with the group effect regressed out. Model features included post-vaccination limma voom-normalised expression (E) values from the top 200 differentially expressed genes (DEGs) identified after the first dose. A similar modelling approach was applied to predict antibody levels following the second vaccine dose, using all DEGs identified post-second dose as input features.
- As you rightly point out, with small sample sizes and a large number of parameters, some degree of overfitting is indeed expected. However, the model is still able to explain 50–60% of the variance in the dependent variable, and the predicted values are significantly correlated with the observed values. This indicates that, despite potential overfitting, the model retains meaningful predictive power and provides useful insights. To highlight this consideration to the readership we have include the below in the discussion/limitation section.
 - Page 16; line 35-41— “The modest sample size, coupled with a high number of parameters, likely contributed to some degree of overfitting in the machine learning model, as evidenced by higher predictive accuracy in the training set compared to the test set. Nevertheless, the model was able to explain over 50–60% of the variance in post-vaccination ELISA levels, and the predicted values remained significantly correlated with the observed values. These findings suggest that, despite the potential for overfitting, the model retains substantial predictive value and offers meaningful biological insights.”
- We have also included a justification for the selection of the ML model we chose.
 - Page 23; line 7-11— “A linear SVR was selected due to its robustness in high-dimensional settings, where the number of features exceeds the number of observations. In such contexts, linear models are less prone to overfitting and often perform more reliably. Moreover, SVR with a linear kernel offers greater interpretability compared with many non-linear alternatives, allowing clearer insight into the contribution of individual features to the predicted outcome.”

- We did examine correlations between individual genes and subsequent antibody levels. Although some associations reached statistical significance, the proportion of variance explained at the single-gene level was substantially lower than that achieved by the machine learning model. For transparency and to support reader interest, we have included the correlation results in the supplementary tables.
 - Page 10 line 11-12 — *Correlation coefficients between gene expression and subsequent antibody levels are shown in Supplementary table 6 and Supplementary table 7.*

• Single cell analysis of B cells: In lines 315 - 317: authors state “relative frequency of plasma cells was observed for the majority of vaccinees at 10 days after dose 1 and 7 days after dose 2”, but figure only shows dose 1 data only.

- Figure 6 — The figure shows data after both vaccine doses but to improve clarity I have included in the axis the time point in relation to the vaccine dose (e.g., Dose 2 + 7 days).

In Figures 6d and 6e, the number of plasma and proliferating cells per donor and time point should be reported. Given the low frequency of plasma cells (~0.1% in some samples), providing cell number is important for assessing the reliability of relative frequency estimates.

- Thank you for this comment for clarity we have included supplementary table 8 that detail the cell counts per sample.
 - Page 10 Line 30 — *PCA was performed on the cytometric data (Total of 295,136 CD19+ viable B cells from 11 individuals), cell clusters were labelled as immature, naïve, unswitched memory, switch memory and plasma cells (Figure 6a and Supplementary table 8).*

Line 318 – 319 and Figure 6e: the authors claim that “a nominal statistical significance of decrease in proliferating B cells”, but Figure 6e has 4 groups, it’s not clear which specific groups are being compared.

- Thank you for this helpful observation. We acknowledge that this was not clearly explained in the original manuscript. In this analysis, post-vaccination timepoints were combined to allow comparison of baseline cell proportions against a single aggregated post-vaccination group. This approach was taken due to the limited number of samples available at individual timepoints. We have edited the results text to clarify this point.
 - Page 11 line 8–10 — *In the scRNA-seq data, a nominally significant decrease in proliferating B cells (unadjusted p-value < 0.05) was also observed when comparing baseline samples with aggregated post-vaccination data (Figure 6e).*

Figure 6f, 6g — The figure legend should include the number of samples in each group, statistical comparison method used and indicate whether the samples are paired or not. Also, have the p values been adjusted for multiple testing?

- We have included this information in the figure 6 legend
 - *f) CDRH3 length in IgM plasma cells at study time points (day 1 n=2, day 11 n=6, day 64 n=3, day 92 n=9). g) CDRH3 length in IgG plasma cells at study time points (day 1 n=8, day 11 n=37, day 64 n=7, day 92 n=32).*

• DEG analysis based on pseudo bulk gene expression of plasma cells: As mentioned earlier, plasma cell cluster is rather small. Without cell numbers per donor and time point, it’s hard to estimate reliability of DEG results (with too few cells, the result may be highly sensitive to noisy signals).

- Thank you for this comment for clarity we have include a supplementary table detail the cell counts per sample.
 - *Supplementary table 8: Table of the cell type counts per sample in the single cell RNA-sequencing data*

The figure legend of figure 7 should be clearer. For figure 7b and 7c, what are log2FC cutoffs the authors used? For figure 7d and 7e, how ‘Down’, ‘NS’, and ‘UP’ are defined? If NS means non-significant, why they are shown here? Can authors also show leading edge genes in each pathway (or include in supplementary tables)? The authors may want to remove this section from the manuscript or discuss the limitations.

- For figures 7a and 7a there is no log2FC cutoff the significance reflects FDR p-values < 0.05, we have included this in the figure legend for clarity.
 - *Volcano plot highlighting DEGs (false discovery rate [FDR] <0.05, red upregulated and blue downregulated) in plasma cells 10 days following the first study vaccine.*
- We have removed the NS pathways and included the legend that significance thresholds (i.e., FDR p-values < 0.05)
 - *Gene set enrichment analysis on differentially expressed gene list from pseudobulk plasma cells 10 days following the first study vaccine compared with baseline, the most differentially regulated upregulated and downregulated pathways are displayed (FDR <0.05).*

- We have also included the total list of significantly enriched pathways and the corresponding leading-edge genes in supplementary tables 9 and 10.
 - *Supplementary table 9: Table of significant enriched pathways (FDR <0.05) from gene set enrichment analysis (C5 biological process) of differentially expressed genes in pseudobulk plasma cells at baseline compared with 10 days after the first dose of vaccine.*
 - *Supplementary table 10: Table of significant enriched pathways (FDR <0.05) from gene set enrichment analysis (C5 biological process) of differentially expressed genes in pseudobulk plasma cells at baseline compared with 7 days after the second dose of vaccine.*
- We have also included an additional comment in the limitation section of the discussion
 - Page 16, line 30-34 — *However, the scRNA-seq experiment was limited by the scarcity of plasma cells among the total peripheral blood B cells with only 215 plasma cells being captured in the scRNA-seq experiment. This low number may particularly limit the generalisability of the pseudobulk plasma cell analysis, as the comparison was based on a small number of cells.*

MINOR COMMENTS:

- Figure 2g is cited before Figure 2f.
 - Thank you have moved the sections so 2f is cited first.
- In Fig6c, the gene names are not visible.
 - We have increased the size of the gene names in Fig6C
- Lines 311–313 refer to Supplementary Figures 7a and 7b, but only a single Supplementary Figure 7 is provided
 - Thank you we have corrected this there is indeed only one panel in Figure 7
- Acronyms need to be defined in abstract “EBOV GP-binding antibodies”
 - Thank you we have defined this acronyms — “*Orthoebolavirus zairensis (EBOV) glycoprotein-binding antibodies.*”
- Typo in gene name: “but their impact on CHR3 structure and epitope interactions are not known”.
 - Thank you we have corrected this to “CDRH3”
- Missing word: “Moreover, observed a cluster of highly similar CDRH3 sequences (within one hamming distance)”
 - Thank you we have corrected by inserting “we”
- Page 9 Line 15: “used by the an EBOV GP binding monoclonal that is an edit distance of 2”
 - Thank you we have removed erroneous word “the”

Reviewer #5 (Remarks to the Author):

REVIEWER COMMENTS

Reviewer #4 (Remarks to the Author):

The authors have addressed all comments, and the manuscript is now much clearer and easier to understand.

A few minor typos still exist:

- Line 328 "vaccinees".

Thank you we have edited these sentences to replace the words vaccinees .

"A relative increase in the frequency of plasma cells in peripheral blood was observed for the majority of individuals 10 days after vaccine dose 1 and 7 days after vaccine dose 2 in both the cytometry and scRNA-seq data."

- Fig 6 d and e: It would be helpful to label the groups the individuals are from.

We have included group label in these plots.

Reviewer #5 (Remarks to the Author):
